# A Causal Framework for Making Individualized Treatment Decisions in Oncology

**DOI:** 10.3390/cancers14163923

**Published:** 2022-08-14

**Authors:** Pavlos Msaouel, Juhee Lee, Jose A. Karam, Peter F. Thall

**Affiliations:** 1Department of Genitourinary Medical Oncology, The University of Texas MD Anderson Cancer Center, Houston, TX 77030, USA; 2Department of Translational Molecular Pathology, The University of Texas MD Anderson Cancer Center, Houston, TX 77030, USA; 3David H. Koch Center for Applied Research of Genitourinary Cancers, The University of Texas, MD Anderson Cancer Center, Houston, TX 77030, USA; 4Department of Statistics, University of California, Santa Cruz, CA 95064, USA; 5Department of Urology, University of Texas MD Anderson Cancer Center, Houston, TX 77030, USA; 6Department of Biostatistics, The University of Texas MD Anderson Cancer Center, Houston, TX 77030, USA

**Keywords:** adjuvant therapy, causal diagrams, individualized inferences, patient-specific decision-making, precision medicine, prognostic biomarkers, predictive biomarkers

## Abstract

**Simple Summary:**

Physicians routinely make individualized treatment decisions by accounting for the joint effects of patient prognostic covariates and treatments on clinical outcomes. Ideally, this is performed using historical randomized clinical trial (RCT) data. Randomization ensures that unbiased estimates of causal treatment effect parameters can be obtained from the historical RCT data and used to predict each new patient’s outcome based on the joint effect of their baseline covariates and each treatment being considered. However, this process becomes problematic if a patient seen in the clinic is very different from the patients who were enrolled in the RCT. That is, if a new patient does not satisfy the entry criteria of the RCT, then the patient does not belong to the population represented by the patients who were studied in the RCT. In such settings, it still may be possible to utilize the RCT data to help choose a new patient’s treatment. This may be achieved by combining the RCT data with data from other clinical trials, or possibly preclinical experiments, and using the combined dataset to predict the patient’s expected outcome for each treatment being considered. In such settings, combining data from multiple sources in a way that is statistically reliable is not entirely straightforward, and correctly identifying and estimating the effects of treatments and patient covariates on clinical outcomes can be complex. Causal diagrams provide a rational basis to guide this process. The first step is to construct a causal diagram that reflects the plausible relationships between treatment variables, patient covariates, and clinical outcomes. If the diagram is correct, it can be used to determine what additional data may be needed, how to combine data from multiple sources, how to formulate a statistical model for clinical outcomes as a function of treatment and covariates, and how to compute an unbiased treatment effect estimate for each new patient. We use adjuvant therapy of renal cell carcinoma to illustrate how causal diagrams may be used to guide these steps.

**Abstract:**

We discuss how causal diagrams can be used by clinicians to make better individualized treatment decisions. Causal diagrams can distinguish between settings where clinical decisions can rely on a conventional additive regression model fit to data from a historical randomized clinical trial (RCT) to estimate treatment effects and settings where a different approach is needed. This may be because a new patient does not meet the RCT’s entry criteria, or a treatment’s effect is modified by biomarkers or other variables that act as mediators between treatment and outcome. In some settings, the problem can be addressed simply by including treatment–covariate interaction terms in the statistical regression model used to analyze the RCT dataset. However, if the RCT entry criteria exclude a new patient seen in the clinic, it may be necessary to combine the RCT data with external data from other RCTs, single-arm trials, or preclinical experiments evaluating biological treatment effects. For example, external data may show that treatment effects differ between histological subgroups not recorded in an RCT. A causal diagram may be used to decide whether external observational or experimental data should be obtained and combined with RCT data to compute statistical estimates for making individualized treatment decisions. We use adjuvant treatment of renal cell carcinoma as our motivating example to illustrate how to construct causal diagrams and apply them to guide clinical decisions.

## 1. Introduction

Clinicians routinely make individualized treatment decisions for their patients based on data that may come from multiple sources, including randomized clinical trials (RCTs), observational studies, and preclinical experiments [1,2,3]. Combining such information is a procedure known as *data fusion* [3,4,5,6]. Causal diagrams can be used to facilitate patient-specific decision-making by showing what data should be used and by explicitly representing the assumptions needed to combine data obtained from different sources. To illustrate causal diagrams and data fusion and show how they may be used to guide individualized treatment decisions, we will use adjuvant treatment for patients with renal cell carcinoma (RCC) as our motivating example.

KEYNOTE-564 was a phase 3 RCT that randomized patients with clear cell RCC, the most common RCC histology [7,8], to either adjuvant pembrolizumab or placebo following nephrectomy with curative intent [9,10]. Eligible patients with stage M0, defined as no history of radiologically visible metastases, were classified into either intermediate-to-high or high risk of clear cell RCC recurrence, as shown in Table 1. Eligible patients with metastasis to a distant organ or tissue (stage M1) who underwent nephrectomy and complete resection of all metastatic disease were classified as “M1 with no evidence of disease” (M1 NED) (Table 1). At the scheduled interim analysis after a median follow-up of 24.1 months, based on a stratified Cox proportional-hazards model, the estimated hazard ratio (HR) of pembrolizumab versus placebo for the primary endpoint of disease-free survival (DFS) time was 0.68 with a 95% confidence interval (CI) of 0.53 to 0.87. For overall survival (OS) time, the trial’s key secondary endpoint, the estimated HR was 0.54 with a 95% CI of 0.30 to 0.96. Several prespecified comparisons within subgroups did not refute the assumption that the HR was the same across all subgroups examined. In the pembrolizumab arm, no treatment-related deaths occurred, but grade ≥3 treatment-related adverse events (AEs) occurred in 18.9% of patients, versus 1.2% of placebo patients [9]. These results led to the approval of adjuvant pembrolizumab by the United States Food and Drug Administration (FDA) for patients with RCC. This led to a discussion of how best to choose adjuvant therapies for RCC based on patient-level characteristics [10], which is the focus of the present paper.

Utilizing observational and RCT datasets outside of KEYNOTE-564, fitted statistical regression models have been developed to predict the DFS and OS probabilities of new patients with RCC who did not receive adjuvant pembrolizumab [11]. One such example is the Assure RCC prognostic nomogram, which is a freely available web-based computer program (https://studies.fccc.edu/nomograms/492; accessed on 20 July 2022) that uses fitted statistical regression models to risk stratify patients with RCC based on their age, pathologic tumor size in centimeters (cm), RCC histology, Fuhrman nuclear grade, presence and extent of vascular invasion, and presence or absence of coagulative necrosis, pathological lymph node involvement, and sarcomatoid features [11]. Additionally, comprehensive molecular characterization of clinical tissue samples, as well as experimental preclinical in vitro and in vivo studies, have provided substantial insights into the biology of RCC [12,13,14,15,16,17,18,19]. This biological knowledge can be represented by a causal diagram, which may shed light on the transportability of treatment effect estimates obtained from RCTs to the population of RCC patients seen in the clinic and may facilitate the computation of nomograms for predicting the clinical outcomes of these patients [3,5,20].

To illustrate how causal diagrams and properly formulated statistical regression models can inform patient-specific inferences, we will focus on the clinical decision of whether or not to recommend adjuvant pembrolizumab to patients with RCC based on their individual characteristics. However, the concepts that we will discuss are generalizable to any malignancy, more complex treatment options, and other clinical settings, such as neoadjuvant treatment and metastatic disease.

## 2. Causal Diagrams

It is important to first distinguish between population parameters, a sample of observed data, and statistical estimators. A parameter, which is easily understood but not observed, is a population quantity such as median survival time, response probability, or the effect of a covariate or treatment on a clinical outcome. We denote a clinical outcome by Y, which may be survival time or a response indicator, and write y for an observed value of Y, such as y = 12 months. The same convention will be used for other variables that may take on different values in some random fashion, such as X for treatment and x for a particular treatment x, for example, denoting X = 1 for pembrolizumab and X = 0 for placebo or standard of care. A statistic, which is anything that can be computed from observed data using a formula or algorithm, may be used to estimate a parameter. For example, to estimate a population median, a computed sample median may be 14.5 months with a 95% CI of 9.4 to 18.7, keeping in mind that a second sample would give different numbers because outcomes and samples are random. A Kaplan–Meier plot [21] is a statistical estimator of the survival function of the population represented by the sample used to compute the plot, and here the population survival function is the parameter. The underlying statistical principle is that, ideally, a sample of Y values should be obtained in such a way that it represents the population of interest, and statistical sampling theory provides a wide array of methods for ensuring this representation [22,23,24]. The design of experiments such as clinical trials is distinct from sampling theory and can be facilitated by the explicit representation of the causal relationships we wish to estimate.

Relationships between patient covariates, treatments, and clinical outcomes can be represented by causal diagrams, such as directed acyclic graphs (DAGs) [4,25,26]. The simplest causal relationship occurs when a variable X, known as the “exposure” in epidemiology or “treatment” in clinical medicine, directly transmits its effect on an outcome Y of interest (Figure 1A). In the RCC setting, the treatment, either adjuvant pembrolizumab (X = 1) or placebo (X = 0), acts directly on the time to disease recurrence or death, defined as Y = DFS time (Figure 1B). A prognostic variable Z may also directly affect DFS time (Figure 1C,D), and in most settings, there is a vector of several prognostic variables. A confounder C is a variable that influences both the outcome Y and treatment X (Figure 1E). For example, the presence (C = 1) or absence (C = 0) of sarcomatoid features acts as a confounding variable if it influences both DFS time and whether a physician chooses to treat a patient with adjuvant pembrolizumab (Figure 1F).

Another type of causal relationship occurs when the effect of treatment on the outcome is not entirely direct, but rather is transmitted, at least in part, through a third variable, known as a *mediator*, M (Figure 2A). For example, the effect of adjuvant pembrolizumab compared with placebo on DFS time is mediated by the RCC immune microenvironment (Figure 2B). The immune microenvironment of each RCC histologic subtype responds differently to immune checkpoint therapies, such as pembrolizumab [8], and the microenvironment, in turn, affects DFS time. Consequently, the effect of a given treatment on DFS time should vary with the immune microenvironment of each RCC histologic subtype because the RCC immune microenvironment acts as a mediator.

As additional variables are included, causal diagrams become more complex. For example, we can also include the influence of epithelial-mesenchymal transition (EMT) on sarcomatoid dedifferentiation, as shown in Figure 2C. A more detailed causal diagram may be obtained by including β-catenin levels that causally influence EMT (Figure 2D) [27]. As more variables are added, a causal graph may become too complex to interpret usefully. An important practical goal thus may be to reduce an overly complex causal graph to a more coarse-grained causal description that can be used to deal with the problem at hand [28,29,30,31]. For example, the causal diagram in Figure 2C may be appropriate if a drug that modifies EMT is being studied. Similarly, the more fine-grained causal diagram in Figure 2D may be used in a study evaluating the prognostic role of assays measuring β-catenin levels. Choosing the right level of granularity for a causal diagram is a subjective decision that should be guided by the goal to use the diagram as a practical tool. This can be performed by investigating how causal relationships may change when finer or coarser resolutions are chosen.

### 2.1. Selection Diagrams

An important refinement of a causal diagram is to identify variables by which populations, or studies that are designed to represent them, may differ [5]. This is achieved by including graphical objects called *selection nodes* that, while they are not variables themselves, use arrows to point to variables that have different sets of possible values between populations [5,32]. If an arrow points from a selection node S to a variable *Z* (Figure 3A), this says that the possible values of *Z* differ between populations. If no arrow points to *Z*, then all populations of interest have the same set of possible *Z* values. A third scenario occurs if an arrow points from a variable *Z* to a selection node S (Figure 3B), which indicates the presence of selection bias, wherein there are differences that are not between the populations but instead are between the datasets due to sampling artifacts. Selection nodes with outgoing arrows are not mutually exclusive with selection nodes with incoming arrows, and both types of arrows can appear in the same selection diagram if they represent two distinct features of the data collection process [33,34]. In the present review, we will focus only on transportability scenarios wherein the differences between populations are inherent, which is denoted by selection nodes S having outgoing arrows toward variables (Figure 3A).

A *selection diagram* is a DAG that includes one or more selection nodes. A selection node may be used to show where treatments (Figure 3C) or mediators (Figure 3D) are different across populations. Whether or not a given variable has a selection node can be determined from either external knowledge or the available dataset itself. A selection node identifies a variable in a causal chain that may be affected by study differences. Selection nodes provide an explicit way to determine whether inferences are transportable between populations and, if not, what additional data are needed to ensure transportability [5,32,35,36]. A major practical point is that a selection node may show that the data from an RCT are not representative of a patient seen in the clinic. In this case, either the trial’s results cannot be used to choose that patient’s treatment, or possibly additional data may be obtained and combined with the RCT data to provide a basis for choosing a treatment.

A selection node can be used to show that the effects of treatments (Figure 3C) or mediators (Figure 3D) are different between populations. For example, if an RCT of two RCC treatments is performed in the United States and another RCT of the same treatments is conducted in China, since the US trial will include African Americans and the Chinese trial will not, a selection node should point to the variable “race” (Figure 3E). As another example, if one study includes a treatment given at dose levels 1, 2, and 3 and a second study includes the same treatment at dose levels 3 and 4, then a selection node should point toward treatment (Figure 3C). If, for a targeted biomarker, an “all-comers” population includes patients who were either biomarker positive or negative, while a second study population only includes biomarker-positive patients, then a selection node should point toward the biomarker (Figure 3F). A selection node can indicate that patient age differs between the population of patients being seen in the clinic, which includes patients older than 65 years, and the population of a study that restricted patients to be younger than 65 years (Figure 3G). Because Figure 3G indicates that patient age influences outcome, external data from studies that included patients older than 65 years are needed to obtain valid inferences on the outcome for such patients seen in the clinic. A selection node may also indicate that some studies included patients with sarcomatoid features whereas others did not (Figure 3H), or that a study only included patients with clear cell RCC and no other histologies (Figure 3I). The general practical point is that, if the variables denoted by selection nodes are substantively different than those in a study population, then a conclusion from that study alone cannot be transported to that patient.

In contrast, the range of a variable without a selection node does not change across populations. For example, the lack of a selection node pointing toward “treatment” in Figure 3D indicates that the treatment choices do not vary between populations. Such assumed invariance in specific causal mechanisms allow inferences to be transported from a study population to patients in a different population, e.g., a patient seen in the clinic [5,20,36].

### 2.2. The Do-Calculus

The relationships represented by all types of DAGs, including selection diagrams, can be used to guide a set of mathematical rules known as the *do-calculus* [25,37], which is expressed using conditional probabilities. The central idea is to estimate causal effects by distinguishing between two types of conditional probabilities. The first type conditions on a variable, say X, that simply was observed and thus may have been confounded by other, possibly unknown variables. The second type conditions on a known value x of *X* that was determined by one’s action or intervention, represented by do(X = x), and thus, X is free of confounding effects. The mathematical expression do(X = x) is known as the *do() operator* and is used to denote a known intervention. For example, in an RCT, if a patient with covariates Z and unknown confounders C (Figure 4A) was randomly assigned adjuvant pembrolizumab (X = 1), then the do() operator may be used to replace the conditional probability P^(RCT)^(Y | X, Z) with P^(RCT)^(Y | do(X = 1), Z). This ensures that there are no confounding effects on X, so the causal effect of pembrolizumab versus a control treatment (X = 0) for a patient with covariates Z may be obtained from statistical estimates of P^(RCT)^(Y | do(X = 1), Z) and P^(RCT)^(Y | do(X = 0), Z). These estimates can be computed from the RCT data under an assumed statistical regression model including the effects of X and Z on Y. In contrast, if X is observed and not assigned, then the values of X in a dataset may have been affected by known or unknown confounding variables, which can cause severe bias and invalidate conventional statistical estimators.

In terms of causal diagrams, the expression P^(RCT)^(Y | do(X = x), Z) modifies the graph in Figure 4A by removing all arrows going into X (Figure 4B). This results in a new causal model (Figure 4C) wherein the distribution P^(RCT)^(Y | do(X = x), Z) is the same as P^(RCT)^(Y | X = x, Z). This modification represents the physical effect of the experimental intervention that sets the value X = x, while keeping the rest of the causal model unchanged.

To illustrate how the do() operator works in practice, a common example in the RCC setting is one where a patient in an observational dataset (denoted as OBS), not obtained from an RCT, has X = 1 recorded, indicating that the patient received pembrolizumab, but it is not known how the patient’s physician chose that treatment. In statistical language, treatment selection was biased in the observational dataset. Consequently, in contrast with what is widely believed, fitting a regression model for DFS time Y as a function of treatment X and covariates Z to the observational dataset may not provide an unbiased statistical estimator of the effect of pembrolizumab versus standard of care. From a causal viewpoint, for a patient with covariates Z, the do() operator cannot be applied due to the treatment assignment bias, so while P^(OBS)^(Y | X = x, Z) can be computed, P^(OBS)^(Y | do(X = x), Z) cannot be computed for either X = 0 or X = 1. If, instead, the data came from an RCT of pembrolizumab (X = 1) versus surveillance (X = 0), then one can estimate P^(RCT)^(Y | do(X = x), Z) from the data for each x, since randomization removes any arrows into X. This allows one to obtain an unbiased estimator of the causal effect E(Y | do(X = 1), Z) - E(Y | do(X = 0), Z), which is the difference in expected DFS times for pembrolizumab versus standard of care for a patient with covariates Z. This example illustrates the scientific difference between recording data from medical practice, where physicians choose treatments based on their knowledge and each patient’s baseline information, and data from an RCT, where each patient’s treatment is chosen by randomization in order to obtain a statistically unbiased treatment comparison for the benefit of future patients. That is, data obtained from clinical practice are inherently biased due to each physician using available information to choose the treatment that they think will be best for each patient, while data from an RCT, where each patient’s treatment was chosen by flipping a coin, are unbiased. The point is that randomization is a statistical device to obtain unbiased estimators in order to maximize the benefit for future patients when choosing their treatments.

Three inferential rules, known as the *do-calculus*, have been derived to transform conditional probability expressions involving the do() operator into other, more useful conditional probability expressions. The first rule considers a DAG, such as the one shown in Figure 4D, where we are interested in the probability distribution of the outcome Y following treatment X conditioned on all patient covariates, which we denote by W and C. As noted earlier, in an RCT, which sets do(X = x), all arrows into X are removed (Figure 4E), resulting in the causal model shown in Figure 4F. Rule 1 of the do-calculus guides the insertion and deletion of variables by stipulating that if, after deleting all paths into X, the set of variables C and treatment choice X = x block all the paths from W to Y regardless of the direction of the arrows on these paths (Figure 4F), then P^(RCT)^(Y | do(X = x), C, W) = P^(RCT)^(Y | do(X = x), C). In probability language, the do() operator ensures that Y is *conditionally independent* of the covariates W, given do(X = x) and the covariates C. The practical implication of Rule 1 is that it allows us to simplify our regression model by dropping the covariates W. For example, suppose that C is a prognostic score recorded in an RCT of pembrolizumab versus surveillance, W is the frequency of monitoring for treatment toxicity based on each patient’s prognosis and chosen treatment, and Y is DFS time. The path from W to Y is blocked by conditioning on C, and therefore Y is conditionally independent of W given C and do(X = x). Accordingly, in the RCT of pembrolizumab versus surveillance, we should condition on the prognostic score but not on the frequency of monitoring for treatment toxicity.

Before we move to Rule 2 of the do-calculus, we first need to define *back-door path*. A back-door path is a non-causal path from the treatment X to the outcome Y. As shown in Figure 4G,H, all paths from X to Y that remain after removing all arrows pointing out of X (Figure 4H) are back-door paths because they flow backward out of X into Y through C. Rule 2 of the do-calculus describes a relationship between an outcome Y, an intervention X, and a vector C of covariates by stipulating that, if the covariates in C block all back-door paths from X to Y (as shown for example in Figure 4D,G), then P(Y | do(X = x), C) = P(Y | X = x, C). This rule requires the strong, unverifiable assumption that there are no unknown external confounders that affect both Y and X and do not belong to the observed vector C. Under this assumption, the probability equality says that it is not necessary to randomize patients between X = 1 and X = 0 because knowledge of C allows the do() operator to be applied. The practical implication, however, is not that one may simply fit a regression model for P(Y | X, C) to observational data to estimate the covariate-adjusted treatment effect in an unbiased way. This is because the assumption underlying Rule 2 cannot be verified. In practice, statistical bias correction methods are applied, such as *inverse probability of treatment weighting* (IPTW), described in Section 2.4 below, that use the available covariates in C to provide an approximation to the strong assumption that a vector C of observed covariates blocks all back-door paths from X to Y. Thus, Rule 2 formalizes the motivation for applying statistical methods that correct for bias, or at least mitigate it, when analyzing observational data. Such methods are discussed briefly in Section 3, below.

Rule 3 of the do-calculus guides the insertion and deletion of interventions by stipulating that, if there is no path from X to Y with only forward-directed arrows (Figure 4I), then P(Y | do(X = x), Z) = P(Y | Z). In this case, observing the covariates Z implies that treatment (or exposure) X has no causal effect on outcome Y. An example from epidemiology is the once common belief that soda pop consumption caused poliomyelitis, due to the observation that higher sales of soda pop (X) were positively associated with a higher incidence of poliomyelitis (Y). However, once the season (Z) of the year was recorded, it was seen that both soda pop consumption and poliomyelitis incidence were higher during the summer months (Z = 1) and lower during the winter months (Z = 0). Thus, Y was conditionally independent of X given Z. This phenomenon is commonly described by saying that association does not imply causation, where an apparent effect of X on Y is due to the fact that Z drives both X and Y. In this case, Z is sometimes called a “lurking variable” acting as a confounder, and X is called an “innocent bystander.” For a therapeutic example, suppose that X = 1 corresponds to standard treatment combined with a completely ineffective experimental agent and X = 0 is standard treatment alone, while Z = 1 denotes good prognosis and Z = 0 denotes poor prognosis. If X = 1 is more likely to be given to good-prognosis patients (Z = 1) and less likely to be given to poor-prognosis patients (Z = 0), then the resulting data will show that X and DFS time (Y) are positively associated, with estimates Ê[Y | X = 1] > Ê[Y | X = 0] apparently implying that adding the experimental agent increases expected DFS time. However, once one accounts for prognosis (Z) the estimates are the same, that is, Ê[Y | X = 1, Z] = Ê[Y | X = 0, Z], aside from random variation in the data. This is because P(Y | do(X = x), Z) = p(Y | Z). That is, DFS time (Y) is conditionally independent of the choice of treatment (X) given prognosis (Z). While this example may seem somewhat contrived, the practice of combining an active control with a new experimental agent and cherry-picking patients with a better prognosis for evaluating the combination in a single-arm trial is not uncommon in oncology [38,39].

### 2.3. Causal Hierarchy

Causal relationships represented by DAGs and expressed by conditional probabilities involving the do() operator apply to one domain, such as the population represented by patients enrolled in an RCT. To use a DAG to determine whether causal knowledge can be transported to another domain, such as the population of patients a physician sees in the clinic [5,36,37], it is useful to distinguish between three types of causal problems. These comprise a hierarchy of increasing difficulty, called the *ladder of causation* (Table 2) [40]. The simplest type of causal problem, layer 1, is predicting an event in a population based on association. A conditional probability, such as P(salmonella infection | diarrhea), characterizes association and may be estimated statistically from data on salmonella and diarrhea, but it is inadequate for addressing layer 2 or layer 3 causal problems without additional experimental knowledge or assumptions. Layer 2 involves determining what happens when an intervention, such as an RCT, is performed in a cohort of patients. The do() operator can be used to denote such interventions, and the do-calculus rules can be used to transform expressions by introducing do() operators in layer 2 conditional probabilities. Layer 3 involves choosing between treatments for an individual patient, and this class of queries requires the *potential outcomes* framework, which is described below. We will give many examples of layer 3 problems in Section 5 because a central focus of our paper is how to integrate observational data (layer 1) with experimental data (layer 2) to make individualized treatment decisions (layer 3). Toward this goal, we first need to describe the potential outcomes approach, which is necessary to answer layer 3 questions.

### 2.4. Causal Inference and Potential Outcomes

To explain potential outcomes, we return to the setting where a physician wishes to decide between adjuvant pembrolizumab (X = 1) and surveillance monitoring (X = 0) for a patient with covariates Z. The physician may consider what the patient’s future outcomes would be for each treatment, which we write as Y^X = 1^ and Y^X = 0^. These are called *potential outcomes* [41,42,43] because only one treatment can be given and, thus, only one of Y^X = 1^ or Y^X = 0^ can be observed. This thought experiment corresponds to an imaginary world where one can make two identical copies of each patient, treat one with adjuvant pembrolizumab and the other with surveillance monitoring, and observe both Y^X = 1^ and Y^X = 0^. Writing the expected value of Y for the copy treated with adjuvant pembrolizumab as E(Y^X = 1^ | Z) and for the copy treated with standard of care as E(Y^X = 0^ | Z), the difference
Ψ = E(Y^X = 1^ | Z) − E(Y^X = 0^ | Z) (1)
is the *causal effect* of treating the patient with adjuvant pembrolizumab rather than surveillance monitoring.

Of course, in the real world one cannot make two copies of a patient. Only one of the two treatments can be given to a patient, and thus only one of each patient’s potential outcomes can be observed. If the patient is treated with pembrolizumab, then X = 1, and the potential outcome Y^X = 0^ for the treatment X = 0 is called a *counterfactual*. This thought experiment has a very useful practical application because it provides a basis for computing an unbiased statistical estimator of Ψ using data from either an RCT or an observational study, under some reasonable assumptions. First, thinking about Y^X = 1^ and Y^X = 0^ only makes sense if, for example, a patient actually treated with adjuvant pembrolizumab has an observed outcome equal to the potential outcome for pembrolizumab, that is, if Y = Y^X = 1^ when X = 1. Similarly, if the patient actually receives surveillance monitoring, X = 0, then Y = Y^X = 0^. This is called *consistency* [43,44,45]. Defining the propensity scores as treatment assignment probabilities Pr(X = 1 | Z), both treatments must be possible for each Z, formally 0 < Pr(X = 1 | Z) < 1, called *positivity*. The potential outcomes Y^X = 1^ and Y^X = 0^ also must be conditionally independent of X given Z. Defining the *causal estimator* as
$$\Psi^{\mathrm{causal}}=\frac{Y*\;X}{Pr(X=1\;|\;Z)}-\frac{Y\;*\;\left(1-X\right)}{\;Pr(X=0\;|\;Z)\;\;\;},$$
it follows from a simple probability calculation that E(Ψ^causal^) = Ψ under these assumptions. That is, the causal estimator Ψ^causal^ is an unbiased estimator of the causal effect Ψ. If the causal estimator is computed for each patient in a sample, assuming that the potential outcomes of each patient are not affected by the treatments given to the other patients, known as the *Stable Unit Treatment Value Assumption* (*SUTVA*) [43,44,45], then the sample mean of the individual causal estimates is an unbiased estimator of Ψ, provided that each treatment assignment probability Pr(X = 1 | Z) (propensity score) is known. If the values of Pr(X = 1 | Z) are not known, as in an observational dataset, then they can be estimated from the data by fitting a regression model for Pr(X = 1 | Z). These propensity score estimates quantify how treatments were chosen based on individual patient covariates, and using them to compute Ψ^causal^ produces an approximately unbiased estimator of Ψ. This is called *inverse probability of treatment weighting* (*IPTW*) estimation [46,47,48], a method often used when analyzing observational data. In a fairly randomized trial, Pr(X = 1 | Z) = 1/2 regardless of Z, and some simple algebra shows that the IPTW estimator equals the difference between the sample means for the two treatments. This also holds for unbalanced randomization, say with probabilities 2/3 for the experimental and 1/3 for the control treatment.

Under a statistical regression model for P(Y | X, Z), the causal treatment effect for a patient with covariates Z is defined as the difference
Ψ(Z) = E(Y | do(X = 1), Z) − E(Y | do(X = 0), Z),
and an estimator of Ψ(Z) for each Z may be obtained by fitting the regression model to RCT data. With observational data, IPTW or a variety of other statistical methods for bias correction may be used [46,49,50]. If two or more datasets are combined based on a causal diagram, then appropriate statistical methods for bias correction also must be used [51,52,53,54].

## 3. Causal Modeling of Treatment Effect Heterogeneity

The family of modern statistical regression models is quite rich and may accommodate a wide variety of relationships between treatments, prognostic covariates, and outcomes [45,55,56,57,58]. In practice, a regression model should be chosen to accommodate the data structure at hand, guided by model criticism, also known as goodness-of-fit analyses, which is performed to ensure that the model provides a reasonable fit to the dataset at hand [56,59,60,61,62]. For example, if goodness-of-fit analyses show that the proportional hazards assumption is not valid for a survival-time dataset, then the Cox model should not be used [63]. Bayesian nonparametric (BNP) regression models for P(Y | X, Z) are a family of robust models that can accurately approximate any distribution, due to the property of “full support” [55,64,65]. Moreover, BNP regression models can be used to correct for bias. An example of this was provided by a computer simulation study in the context of estimating mean survival times of several different multicycle adaptive treatment algorithms, known as “dynamic treatment regimes,” for acute leukemia. The simulations showed that the BNP regression model reliably corrects for bias in an observational dataset if the covariates causing the bias are known and available [66]. Since details of BNP models and the array of statistical methods for goodness-of-fit analyses are beyond the scope of this paper, for simplicity, we will assume that model criticism has been performed, and that the regression model includes a linear component, LIN, that is a parametric function of treatment X and a vector Z of patient covariates. For example, if survival time (Y) follows a log-normal distribution, then LIN = E[log(Y) | X, Z] and a larger LIN is more desirable because it corresponds to a longer mean survival time. If, instead, Y follows a Weibull distribution, then LIN is a component of the log hazard function and a smaller LIN is more desirable because it corresponds to a lower risk of death.

### 3.1. Causal Diagrams and Interaction Parameters

The magnitude and direction of a treatment effect can vary depending on the characteristics of each individual patient. This is called *heterogeneity of treatment effect* (HTE). For example, the effect of adjuvant pembrolizumab on DFS is expected to be different for patients with clear cell RCC compared to patients with other RCC histologies. The Predictive Approaches to Treatment effect Heterogeneity (PATH) consensus statement was developed to provide physicians with guidance on how to account statistically for HTE when computing patient-specific treatment effect estimates [67]. The PATH framework distinguishes between additive (*risk modeling*) and interactive (*effect modeling*) forms of LIN in regression models for computing these estimates [67,68]. To guide this choice, a causal diagram may be used to decide whether a regression model should take the additive form or interactive form. Let Z = (Z_1_,…,Z_p_) denote a vector of patient covariates, and a = (a_1_,…,a_p_) and c = (c_1_,…,c_p_) denote two corresponding vectors of parameters, and write the linear combinations
aZ = a_1_Z_1_ + … +a_p_ Z_p_ and cZ = c_1_Z_1_ + … +c_p_ Z_p_.

The additive (risk modeling) form of a treatment effect has linear term
LIN = b_0_ + b_1_X + aZ,
where b_1_ is the experimental-versus-control treatment effect regardless of Z, while the interactive (effect modeling) form is given by
LIN = b_0_ + b_1_X + aZ + (cZ) X.

In this interactive model, the experimental-versus-control treatment effect depends on the parameters b_1_ and c, and the patient covariates Z.

Causal diagrams can be used to represent additive cases (Figure 5A,B) or interactive cases (Figure 5C,D), allowing users to determine which of the two modeling approaches is more biologically plausible. Figure 5A represents a scenario wherein the treatment choice X and patient covariates Z independently influence the outcome Y, which corresponds to the additive regression model. In an additive risk model that includes only the main effects, b_1_ X + aZ, the covariates Z often are called *prognostic* variables, corresponding to Z in the causal diagram of Figure 5A. Figure 5C represents a scenario where the treatment choice X and a single baseline biomarker B interact by jointly influencing the mediator M that transmits the effect of X on the outcome Y. Here the biomarker B plays the role of Z, and this corresponds to the interactive form, LIN = b_0_ + b_1_X + a B + c BX. The covariate B shown in Figure 5C is called a *predictive* variable because it has a multiplicative interaction effect [2]. The terms prognostic and predictive can be misleading, however, because, as described above, prognostic variables can be used to estimate and predict patient-specific outcomes [2,69]. For the purposes of the present paper, we will use these terms due to their widespread use in the oncology literature [70].

In the RCC clinical scenario illustrated in Figure 5B, the relative treatment effect, typically measured on the HR scale, is stable and independent of the effect of prognostic variables on the DFS outcome. The assumption that the relative treatment effect is stable across different patients is called *treatment effect homogeneity* and is a fundamentally scale-dependent concept, i.e., the estimated relative treatment effect may be stable using one scale but variable using another scale [68,71]. For example, even in cases where the relative treatment effect measured on the HR scale is the same between patients, the treatment’s effect on median survival or absolute risk reduction probabilities can vary widely between patients with different covariates [2,72]. Thus, a more accurate term might be *treatment effect homogeneity in the scale of interest* [73]. Ongoing methodological research is investigating situations with *treatment effect homogeneity in distribution* wherein treatment effect homogeneity is noted across all standard scales [71,73]. In the present paper, for simplicity, particularly in the illustrative examples discussed in Section 5, causal diagrams such as those shown in Figure 5A,B will be used to represent treatment effect homogeneity on the HR scale, which is the most commonly used scale for survival analyses in medicine [74].

In some scenarios, the assumption of treatment effect homogeneity across patient populations may be implausible based on contextual biological knowledge. This is the situation represented by Figure 5C, where a biomarker B influences the relative treatment effect. For example, the HR for the DFS outcome of pembrolizumab compared with placebo may be different depending on RCC histology (B) (Figure 5D). The magnitude and direction of the interactive histology–treatment effect b_3_BX depend on the value of B for each patient, so different values of B may lead to different treatment choices. For example, when analyzing data from an RCT of a targeted experimental treatment represented by X = 1, suppose that it is desired to estimate the effect of a biomarker, represented by an indicator variable B = 1 for biomarker positive and B = 0 for biomarker negative, on overall survival time (Y). In this case, one should assume the interactive form for LIN, where b_2_ is the biomarker’s main effect and b_3_ is the additional treatment–biomarker interaction. The overall experimental treatment effect is b_1_ + b_2_ + b_3_ in biomarker-positive patients and b_1_ in biomarker-negative patients, so the experimental-versus-control treatment effects are (b_1_ + b_2_ + b_3_) − b_2_ = b_1_ + b_3_ if B = 1 and b_1_ − 0 = b_1_ if B = 0. Accordingly, the effect of adjuvant pembrolizumab on DFS is different for patients with clear cell RCC (B = 1) versus other RCC histologies (B = 0), so here RCC histology plays the role of the biomarker. Once a dataset is analyzed under an interaction model, the magnitudes and variances of the parameter estimates play critical roles in deciding whether there are substantively meaningful estimated experimental-versus-control effects in biomarker-positive patients, in biomarker-negative patients, or overall. This is addressed statistically by estimating survival probabilities or performing hypothesis tests within the biomarker subgroups, based on the fitted regression model [75,76].

The example in Figure 5D illustrates that, in many settings, the strong assumption that there is one treatment effect, possibly defined as an HR, that is identical across all patient subgroups may be clinically implausible or not supported by the data. In such cases, the PATH statement recommends using models that include treatment–subgroup interactions to account for HTE [67,68], if the subgroups are known. HTEs may take much more complex forms than the treatment–covariate interactions in the LIN given above. In such a case, more flexible statistical methods such as a regression tree [77,78], neural net [79], or BNP regression model [64] may be applied to capture complicated patterns, such as high-order interactions among X and elements of a vector (Z, B) of variables that includes both patient prognostic covariates Z and a biomarker B. These models lack a uniform component that can be called LIN. A fitted regression model still can be used to compute expected or predicted values of Y, along with their distributions, for a given patient’s (Z, B) = (z, b) and each treatment X = x, in order to choose an optimal treatment for that patient. This was illustrated by a regression analysis of an observational dataset including 151 acute leukemia patients who underwent allogeneic stem cell transplantation (allosct) [66]. A BNP model was fit for Y = log survival time as a function of X = the delivered dose of intravenous busulfan in the preparative regimen for allosct, Age, and the indicator AD of whether the patient had active disease (AD = 1) or not (AD = 0) at transplant. The fitted BNP model showed that the intravenous busulfan dose that maximized mean survival time E(Y | X, Age, AD) varied substantively and nonlinearly with both Age and AD, illustrated graphically by Figure 6 and Figure 7 of Xu et al. [66]. The three-way (X, Age, AD) interaction identified by the BNP model can be used to guide the choice of X by a physician based on the joint causal effect on expected survival time. This would not have been detected by a conventional survival time regression model that included only conventional AUC-Age and AUC-AD interaction terms.

To summarize, there are several objects and actions to consider when using causal ideas to guide one’s choice of data, statistical regression models, and inferential methods to choose an individualized treatment. The first is the patient’s covariates, which may include prognostic variables Z and biomarkers B, the possible treatments X, and the outcome Y that one wishes to optimize. Next, based on clinical and biological knowledge, a causal diagram that may include selection nodes should be constructed. Using the diagram, it should be decided whether the currently available dataset is adequate, or if additional external data or knowledge are needed to remove selection nodes. If necessary, an appropriate statistical method for properly combining data from multiple sources should be applied. The next step is considering possible statistical regression models for P(Y| X, Z, B), performing goodness-of-fit analyses to choose an appropriate model, and fitting it to the data. The causal graphs can help guide these choices, including the decision of whether treatment–covariate interactions should be included in a model’s linear term, if it has one, to account for HTE. If the fitted model shows that there is HTE, then, using each patient’s (Z,B), the estimate of P(Y| X, Z = z, B = b) based on the fitted model for each possible treatment X = x can be used to choose the best x for that patient.

When constructing a causal diagram, it is useful to distinguish between settings where an external dataset is available and may be combined formally with RCT data and settings where basic biological knowledge reliably motivates the belief that a particular agent may only have an anti-disease effect in a particular subpopulation. This is the case, for example, if it is well understood that a targeted agent can only affect a biologically targeted biomarker B, such as a specific histology, cancer cell surface marker, or signaling pathway, that activates the mediator pathway M (Figure 5C), and it is impossible to have a direct effect X → Y without the mediator. In such settings, it makes no sense to include patients who are biomarker negative in an RCT. An example of this will be given in Section 3.4, where the development of the targeted agent trastuzumab for patients with human epidermal growth factor receptor 2 (HER2)-positive breast cancer is discussed.

When making such a determination, however, it must be kept in mind that some novel agents have multiple effects, and an agent may improve patient outcomes by acting on an unknown or unidentified biomarker that is different from the particular biomarker B that has been studied experimentally. In this case, excluding putatively biomarker-negative patients is a mistake, since it deprives them of the new agent’s unanticipated anti-disease effect. Examples of unexpected biological effects include the accidental discoveries of penicillin as an antibiotic [80], sildenafil citrate as a treatment for erectile dysfunction [81], and nitrogen mustard as an alkylating anticancer agent [82]. A recent example of an agent showing efficacy in biomarker-negative patients is the phase 3 testing of the new HER2-targeting antibody-drug conjugate trastuzumab deruxtecan, which substantially improved outcomes in patients with breast cancer whose HER2 biomarker levels were too low to justify treatment with conventional trastuzumab [83].

### 3.2. Transporting Information across Domains: General Principles

The process of combining data from one or more sources to make inferences about treatment effects for a patient with covariates Z in the clinic relies on the statistical idea that there is a population represented by the study data, and a second population to which the patient belongs. In the causal analysis framework, populations are called domains, and whether estimates of causal effects from one domain can be used to make causal inferences about another domain is called *transportability* [5,32,37]. Central issues are whether or not the inference obtained from the study can be transported to make a treatment decision for the patient in the clinic and, if not, what external information can be used to make the inference from the RCT transportable to the patient. The domain D^(RCT)^ represents the study population of an RCT comparing an experimental treatment (X = 1) and a control (X = 0). Causal diagrams are used to determine whether it is necessary to combine external data (denoted by EXT), obtained from observational or experimental studies or possibly other RCTs, with the original RCT data in order to make the inference from the RCT transportable to patients in the domain D^(CLIN)^ in the clinic. A major practical problem is that clinical trial eligibility criteria often restrict D^(RCT)^ so severely that it does not include a substantial portion of D^(CLIN)^, and consequently the RCT data cannot be used to make inferences for many patients seen in the clinic. Thus, the process requires rigorous causal, statistical, and medical thinking.

For individualized treatment decision making, the conditional distributions of P^(CLIN)^(Y | do(X = x), Z) for each possible treatment x are of central interest for making clinical decisions for an individual patient with covariates Z. Practical examples will be described in Section 5, below. These conditional distributions provide a basis for estimating the causal effect Ψ^(CLIN)^(Z) = P^(CLIN)^(Y | do(X = 1), Z) − P^(CLIN)^(Y | do(X = 0), Z) in Equation (1), and thus they answer layer 3 queries expressed using the potential outcomes framework. To obtain estimates of Ψ^(CLIN)^(Z), one may use a causal diagram and a statistical model and combine estimates from different sources, such as ψ^ from D^(RCT)^ and P^(EXT)^(Y | Z) from D^(EXT)^.

### 3.3. Transporting Information across Domains: Additive Models

Assuming the causal relationships shown in Figure 6A, we can use the do-calculus syntax to estimate the distribution of the outcome Y given that treatment do(X = x) is administered to a patient seen in the clinic. The survival time distribution of a patient with covariates Z treated with X = x in an RCT is P^(RCT)^(Y | do(x), Z) since randomization ensures that the do() operator can be applied to remove any arrows that might point to X. The parameters in the regression model are estimated using the RCT data, and the fitted model provides a causal basis for subsequently choosing a treatment for a patient in the clinic with the given covariates. Figure 6B illustrates the clinical scenario where adjuvant pembrolizumab is considered for a patient with RCC based on the results of the KEYNOTE-564 RCT. The advantage of using selection diagrams such as those in Figure 6A,B to determine whether knowledge from different domains can be transported to the domain of patients a physician sees in the clinic (D^(CLIN)^) is a graphical criterion called *S-admissibility* [84]. Denoting all selection nodes in a causal diagram by S and the set of baseline (pre-treatment) covariates by Z, S-admissibility is defined mathematically using the do-calculus syntax as:P (Y | do(X = x), Z) = P (Y | do(X = x), Z, S).

This formalizes how the selection nodes S influence the outcome Y. The S-admissibility criterion states that, if we remove all incoming arrows toward the treatment X from the selection diagram and statistically account for all values of Z across populations, then we can transport knowledge from D^(RCT)^ to D^(CLIN)^ by using the distribution of the covariates Z. Intuitively, this means that a treatment effect can be transported across populations if we can model all relevant mechanisms that influence the outcome of interest. Figure 6A,B show S-admissible examples where we need to model the effect of Z on Y, i.e., P(Y | Z), across populations (Figure 6A) or the effect of prognostic variables on DFS (Figure 6B) to obtain P (Y | do(X = x), Z).

To date, the dataset in KEYNOTE-564 has not been used to generate prognostic risk tools to estimate DFS or OS time distributions of patients with RCC in the adjuvant setting. The Assure RCC prognostic nomogram is an externally developed tool for this purpose using data from the ECOG-ACRIN 2805 (Assure) RCT, which evaluated the efficacy of the tyrosine kinase inhibitors sunitinib or sorafenib versus placebo as adjuvant therapies for RCC [11,85]. The prospective data from this RCT were used, as suggested by the PATH statement [67], to generate a prognostic nomogram that does not include a term for treatment assignment between sunitinib, sorafenib, or placebo [11]. The advantage of using highly annotated prospective rather than retrospective data in observational data analyses is that retrospective data collection is more prone to biases from many sources, including variations in data collection and reporting [86]. The use of such external data to estimate the effect of Z on Y when no treatment is given, i.e., P^(EXT)^(Y | do(X = 0), Z), is quite common [2,68]. These external datasets take advantage of their large sample size and representativeness to provide external validity. In contrast, RCTs often have restrictive entry criteria in order to achieve high internal validity when estimating the effect of each do(x) on Y [3,87,88,89]. Knowledge gained from external and RCT studies may be considered complementary, and both may be needed for making reliable patient-specific decisions for the patients in D^(CLIN)^ when the assumptions shown in Figure 6A hold.

Figure 6C illustrates an example where a confounding variable, C, may influence both Z and Y, and this confounder varies across populations, as indicated by the selection node S2. In this scenario, accounting only for Z is not S-admissible and combining causal information from D^(RCT)^ and D^(EXT)^ cannot be legitimized unless we also weight for C to shield the outcome Y from the source of disparity S2 [37]. A corresponding clinical example is shown in Figure 6D, where GL = geographic location may act as a confounder because it influences both baseline prognostic variables and survival outcomes of patients with RCC treated with adjuvant pembrolizumab versus placebo [90]. In this scenario, if each patient’s GL is known, then a regression model may be parameterized to account for GL, which improves the external validity of prognostic nomograms for RCC recurrence and legitimizes their use on a global scale. This is consistent with the literature on the value of causal knowledge in developing clinical risk prediction models and transporting them across different populations [20,91,92]. In any case, when transporting knowledge from different sources to a target population, one must make the strong assumption that there are no unknown external variables that cause confounding between Z and Y since such a confounder can invalidate the estimate of the effect of treatment X on Y.

In general, the S-admissibility criterion shows what data are needed to estimate P(Y | do(X=x), Z) in a particular domain, such as the domain D^(CLIN)^, which refers to the population of patients seen in the clinic. Whether or not the data needed are available shows whether there is sufficient information for making clinical decisions for the patients in D^(CLIN)^. If we cannot model all relevant mechanisms that influence the outcome of interest using available data, then there is not enough information to make well-informed decisions between treatments X = 0 and X = 1 in the clinic.

### 3.4. Transporting Information across Domains: Interactive Models

Most regression models used to analyze the primary endpoints of RCTs do not include interaction terms and are formulated assuming that no patient subgroups will have an interactive effect with treatment. As noted earlier, more elaborate statistical models are available that facilitate data-driven effect modeling of HTE [93,94,95,96]. Forest plots are a crude graphical approach to represent point estimates and confidence intervals for each of many selected subgroups, so called because such a graph resembles a forest of horizontal lines [2,97,98]. Although originally developed for meta-analysis of multiple RCTs, forest plots are used increasingly for exploratory analyses to search for signals of treatment interactions with specified subgroups, using data from one RCT [98,99]. Because accurately estimating treatment–subgroup interactions requires larger sample sizes than needed to estimate the main treatment effect of an RCT [56,100], many of the computed confidence intervals in forest plots are quite wide [99]. For example, all forest plot subgroup comparisons looking for outcome differences on the HR scale in KEYNOTE-564 were inconclusive at the 0.05 significance level [9]. Using this *p* value cutoff for each of many tests is very misleading, however, since the overall type I error rate is much larger than 0.05 due to multiple testing. This severely limits the usefulness and interpretability of forest plots, since the type I error rates of each individual subgroup comparison should be adjusted downward to maintain an overall type I error rate of 0.05. This adjustment produces wider confidence intervals in the forest plot. For a large number of subgroup comparisons, the adjusted type I error rates should be very small. It is well known that if one examines enough subgroups in an RCT dataset using the type I error rate of 0.05 for each subgroup, then seemingly “statistically significant” treatment–subgroup interactions will emerge even when such interactions do not exist [2,101,102,103]. The process of testing for treatment effects in many selected subgroups, without any scientific rationale for why such effects should exist, is quite common in the published medical literature and often is referred to as *data dredging*. For these reasons, efforts to identify treatment–subgroup interactions should be based on a pre-existing biological or clinical rationale for how particular subgroups, such as those defined by biomarkers, may mediate the effect of treatment [2,67,68,101,102,103].

We illustrate how causal diagrams and effect modification can be informed by biological knowledge derived from preclinical experiments using HER2 signaling in breast cancer. This is one of the most well-established predictive biomarkers in oncology [104,105]. Analyses of observational data on tissues from breast cancer patients showed that the gene encoding HER2 is amplified in 25% to 30% of breast cancers [106,107]. Studies in cell line and animal models showed that *HER2* amplification, identified as a biomarker B, drives the growth of breast cancer cells via the upregulation of oncogenic HER2 signaling, which acts as a mediator M [104,108,109,110,111]. This motivated the development of the monoclonal antibody trastuzumab, which is targeted to inhibit this signaling in *HER2*-amplified breast cancer [112]. The key point is that, if the biomarker variable B indicates cell surface upregulation of oncogenic HER2 signaling M after treatment, then trastuzumab can only affect a clinical outcome Y by acting on M. This is the same causal path as that expressed in Figure 5C, whereby the upregulation of the biomarker B activates the pathway M that mediates the effect of trastuzumab on Y. For simplicity, denote the value of the upregulation biomarker recorded at enrollment by B = 0 for absence or 1 for presence. This motivated designers of the pivotal phase 3 RCT that led to the approval of trastuzumab for the treatment of metastatic breast cancer to only allow enrollment of patients with baseline *HER2* amplification (B = 1), which is called *enrichment*, and to test the efficacy of combining trastuzumab with chemotherapy versus chemotherapy alone in those patients [113]. If an all-comers trial had been conducted, also including metastatic breast cancer patients without baseline *HER2* amplification (B = 0), whose tumors were not driven by oncogenic HER2 signaling, it would have exposed those patients to harm from trastuzumab toxicity without any chance of benefit. Additionally, because only 25% to 30% of patients would have been expected to respond to trastuzumab, a hypothetical HER2-naïve trial utilizing an interactive effect modeling approach would require a substantially larger sample size to detect trastuzumab efficacy for each of the patient subgroups, with and without HER2 amplification. The biologically informed enrichment strategy in the actual trial provided a reliable estimate of the main effect of trastuzumab in patients with *HER2* amplification and facilitated the identification of a treatment interaction signal between the degree of *HER2* overexpression and trastuzumab efficacy [111]. Figure 7A illustrates the biological assumptions underlying the RCT, where trastuzumab acts on oncogenic HER2 signaling to impact the survival outcome. The selection node S shows that oncogenic HER2 signaling differs between the patient population enrolled in the RCT, which has *HER2* amplifications, and the population of patients seen in the clinic (domain D^(CLIN)^), which may or may not have *HER2* amplifications. The general form of the diagram is shown in Figure 7B, where the effect of a treatment X on outcome Y is mediated by a variable M influenced by a biomarker B that differs between patient subpopulations. This causal framework for biological treatment effect modification subsumes the approach of the PATH statement, which focuses exclusively on identifying treatment–subgroup interactions without incorporating prior causal knowledge from external sources, such as preclinical experimental studies [67,68]. Furthermore, as we will show below, similar to the approach under the additive modeling framework, the structural causal relationships represented in Figure 7B allow us to determine whether inferences can be transported from different domains to the domain D^(CLIN)^ of patients seen in the clinic.

We next provide a causal rationale for enrichment in any setting similar to that of the trastuzumab trial, which only enrolled patients with a biomarker for *HER2* overexpression. Suppose that preclinical data show on fundamental grounds that a new targeted agent (X = 1) can only affect Y through M as a mediator pathway influenced by a biomarker B. First, as a comparator, consider an all-comers RCT, and denote the value of B recorded at enrollment by B = 0 or 1. This trial’s data can be used, as usual, to estimate the effects of X on Y by including the terms b_1_X + b_2_B + b_3_XB in the statistical regression model’s linear term. The following considerations provide a rationale for why an enrichment trial enrolling only patients with B = 1 should be conducted instead.

Suppose that experimental studies to determine the effect of X via a mediator M influenced by biomarker B are conducted in an external domain D^(EXT)^ that includes D^(CLIN)^. The experimental studies of M can include preclinical research, as was performed with trastuzumab, or additional RCTs focused on specific biological contexts. The first requirement is that preclinical laboratory experiments, with domain D^(LAB)^, must be designed with a randomized 2 × 2 factorial form including all four combinations of B = 0 or 1 and X = 0 or 1. This ensures that these preclinical experiments can show that the targeted agent and control treatment have identical effects on all outcomes when the biomarker is absent. This is expressed by the do-calculus equation
P^(LAB)^(Y^(LAB)^ | do(X = 1), B = 0) = P^(LAB)^(Y^(LAB)^ | do(X = 0), B = 0),(2)
where Y^(LAB)^ represents a preclinical laboratory experiment outcome variable. This ensures that excluding biomarker-negative patients from the planned RCT will not deprive them of an unanticipated anti-disease effect from the targeted agent.

Based on the selection diagram shown in Figure 7B, modeling the effect of the covariate B on the outcome Y fulfills the S-admissibility criterion. While an all-comers RCT would provide estimators of P^(RCT)^(Y | do(X = x), B = b) for all four combinations of x = 0 or 1 and b = 0 or 1, the following causal argument implies that an RCT with enrollment restricted to patients with B = 1 can be conducted and an estimator of P^(RCT)^(Y | do(X = x), B = b) can still be obtained.

Under some key assumptions, data from the three domains D^(CLIN)^, D^(RCT)^, and D^(EXT)^ can be used to estimate the probability distribution P^(CLIN)^ of the outcome Y, given that a treatment X = 1 or X = 0 is administered to a patient seen in the clinic. The first step is to use the results of a 2 × 2 factorial laboratory experiment based on the strong assumption that equation (2) with preclinical outcome Y^(LAB)^ in laboratory animals implies
P^(RCT)^(Y | do(X = 1), B = 0) = P^(RCT)^(Y | do(X = 0), B = 0)
for the corresponding clinical outcome Y in the RCT in humans. This assumption says that, for biomarker-negative patients, the preclinical result that there is no difference between the effects of the targeted agent and the control on Y^(LAB)^ in animals with B = 0 implies that this also holds for the clinical outcome Y in humans with B = 0. Since an RCT that does not include biomarker-negative patients does not provide data for estimating E(Y | do(X = x), B = 0) for either x = 0 or 1, an estimate for x = 0 might be obtained using external data from another human trial, either randomized or single-arm, that included the control treatment x = 0 and evaluated the biomarker for each patient at enrollment. This external data for biomarker-negative patients would provide an estimate of P^(EXT)^(Y | do(X = 0), B = 0), which then might be used to estimate P^(RCT)^(Y | do(X = 0), B = 0), with appropriate statistical correction for bias due to between-study effects. Next, the “mouse-to-man” assumption that biological knowledge from preclinical experiments in mice implies that treatments X = 1 and X = 0 must have the same effect in biomarker-negative humans in turn implies that the estimate of P^(EXT)^(Y | do(X = 0), B = 0) also can be used to estimate P^(RCT)^(Y | do(X = 1), B = 0). Finally, this inferential process may be elaborated by including additional covariates Z additively in the regression model to help personalize treatment decisions and improve precision, which will be further discussed in Section 3.5. The enriched RCT data thus can be used to estimate P^(RCT)^(Y | do(X = x), B = 1, Z) for both X = 0 and X = 1 as the basis for making clinical decisions in biomarker-positive patients while also accounting for the effects of Z.

Figure 7C illustrates a clinical scenario in which adjuvant pembrolizumab is being considered based on the results of the KEYNOTE-564 RCT for a patient with a different RCC histology, and thus a different effect of pembrolizumab (X = 1) on the post-treatment tumor immune microenvironment status M [8], rather than the clear cell RCC subtype that was exclusively enrolled in KEYNOTE-564. For simplicity, we will assume that all datasets in domain D^(LAB)^ are experimental. While in Figure 6A we fused experimental with observational data from D^(EXT)^, the information sources used in Figure 7B can be a mixture of the experimental results from D^(RCT)^ and the preclinical experimental results in D^(LAB)^.

Notably, the S-admissibility criterion would be satisfied even in the scenario where *HER2* amplification status acts as both a prognostic and predictive biomarker for overall survival time [114]. As shown in Figure 7D, covariate weighting for HER2 amplification status would block both the predictive biomarker path toward oncogenic HER2 signaling and the prognostic direct path toward overall survival. Intuitively, this means that if we have enough data to estimate the predictive effect of *HER2* amplification status (i.e., the interactive term of treatment × *HER2* amplification status), then we have enough data to also account for the prognostic additive effect of *HER2* amplification status on overall survival time.

### 3.5. Transporting Information across Domains: General Framework

The causal relationships illustrated in Figure 6A and Figure 7B inform the study designs needed to predict outcomes for patients seen in the clinical domain, D^(CLIN)^, under the additive effect model (Figure 6A) and the biological treatment effect modification (Figure 7B) frameworks. In practice, clinicians should always consider both an additive effect model and the possibility of biological treatment effect modification when making patient-specific decisions involving knowledge transferred from different domains to D^(CLIN)^. This corresponds to the structural causal relationships represented in Figure 8A. Using the causal relationships of Figure 8A, assume that an RCT of treatment X is conducted in the domain D^(RCT)^, observational studies representative of all baseline prognostic variables Z are conducted in the domain D^(EXT)^, and experimental studies to estimate the influence of a biomarker (represented by B at baseline) on the mediating pathway M for the effect of X are conducted in domain D^(LAB)^, to yield P^LAB^(M | (do(X = x), B). The S-admissibility criterion would be fulfilled because with this information we can model the effects of both B and Z on the outcome Y.

Figure 8B illustrates the corresponding clinical scenario where adjuvant pembrolizumab is considered based on the results of the KEYNOTE-564 RCT for patients who may have a prognostic risk or RCC histology different from those of patients enrolled in KEYNOTE-564. Using this diagram, we can determine whether we possess sufficient knowledge to make such cross-population predictions for patients seen in clinical practice. Note also that the S-admissibility criterion would be fulfilled even in more complex scenarios whereby RCC histology also directly influences DFS (Figure 8C) and the prognostic variables (Figure 8D).

For individual patients seen in the clinic, the structural causal model represented by Figure 8A corresponds to a potential outcomes formula that conditions both the patient’s prognostic risk factors Z and the biological effect modifier B to determine the individualized causal treatment effect,
Ψ (Z, B) = E(Y^X = 1^ | Z, B) – E(Y^X = 0^ | Z, B).(3)

This general formula is reduced to the additive, effect modeling Equation (1) in patients for whom a treatment effect modification is not expected biologically.

### 3.6. Transporting Information across Domains: More Complex Scenarios

The S-admissibility criterion can help us determine algorithmically whether we possess sufficient knowledge to make decisions for patients seen in the clinic in scenarios that are more complex than the adjuvant pembrolizumab scenarios we are focusing on in this paper. For example, as illustrated in Figure 9A, the effect of the treatment choice X on the outcome Y may be mediated by two distinct mediators, M1 and M2, which are influenced by respective biomarkers B1 and B2, which also directly influence the treatment choice X. Furthermore, prognostic variables Z may also independently influence the outcome Y. The populations of patients studied may differ in B1 (denoted by the selection node S1), B2 (denoted by S2), and Z (denoted by S3). In this situation, after removing all arrows pointing into X (Figure 9B), the targeted quantity Y can only be separated from S1, S2, and S3 if we statistically weight for B1, B2, and Z. Accounting for each of these variables alone will not be enough. After obtaining the necessary knowledge from each available domain, we can use the potential outcomes formula: Ψ (Z, B1, B2) = E(Y^X = 1^ | Z, B1, B2) – E(Y^X = 0^ | Z, B1, B2) to determine the expected individualized causal treatment effect for each patient seen in the clinic. 

A corresponding clinical scenario is shown in Figure 9C, wherein a clinician is called to choose between either immunotherapy or anti-angiogenic targeted therapy for patients seen in the clinic with metastatic RCC. The effect of choosing between immunotherapy or anti-angiogenic targeted therapy is mediated by the RCC immune microenvironment, which is influenced by the baseline immune signature of the RCC tumors. Furthermore, the effect of choosing between the two treatments is also mediated by the RCC angiogenic microenvironment, which is influenced by the baseline angiogenic signature of the RCC tumors. The immune and angiogenic signatures also influence treatment choice in the studied populations. In addition, the survival outcome is independently influenced by baseline prognostic covariates. As shown in Figure 9D, after removing all arrows pointing into the variable “treatment choice,” the pathways from S1, S2, and S3 toward the survival outcome can only be blocked if we covariate weight for shifts in prognostic variables and in immune and angiogenic signatures.

## 4. Treatment Effect Calculator

To illustrate the clinical utility of the causal relationships expressed in Figure 8, along with the potential outcomes formula of Equation (3), we provide a simple calculator (Appendix A) that can be used by clinicians to estimate survival probabilities when choosing between treatments. The calculator assumes an exponential distribution for survival times, which implies that the hazard of death for each patient is constant over time. This is the simplest time-to-event distribution for modeling survival times [115,116,117], and it is also the distribution implicitly assumed by the statistical models used for the primary and key secondary endpoint analyses of KEYNOTE-564 [9]. More complex regression models with less restrictive assumptions can be used, as appropriate, in scenarios where the exponential distribution does not provide an adequate fit to the data [118,119]. These models require specialized but widely available statistical software to perform data analyses.

An exponential distribution for Y = DFS time is characterized by one parameter, its hazard (failure rate) parameter, h [2]. The mean DFS time is μ = 1/h, and median DFS time is *ln*(2) μ, where *ln* denotes the natural logarithm. The probability of DFS beyond a given time t is P(Y > t) = e^−ht^, where e = 2.718 is Euler’s number, so *ln*(e) = 1. For example, if the hazard rate is h = 0.04 per month, then mean survival time is μ = 25 months, median survival time is 17.3 months, and the 12-month survival probability is P(Y > 12) = e^−(0.04 · 12)^ = 0.619. In most settings, the hazard rate varies with a patient’s prognosis Z, treatment X, and possibly a treatment effect modifier, such as a biomarker B. This may be accommodated by an exponential regression model, where, for example, an additive model’s hazard function might take the form h(X, Z, B) = exp(LIN) with LIN = b_0_ + b_1_X + b_2_Z + b_3_B, and treatment–covariate interaction terms may be added to LIN as appropriate. The experimental-versus-control HR for given Z and B is
HR(Z, B) = h(1, Z, B)/h(0, Z, B).

Under the above additive model, HR(Z, B) = exp(b_1_) for all Z and B, and it is the treatment effect on h for all patients. This is the same HR as that under the simplest exponential model without effects for Z or B, so if one HR is reported then it may be unclear whether an additive regression model was assumed. Alternatively, for example, the interactive model with
LIN = b_0_ + b_1_X + b_2_Z+ b_3_B + b_4_ XB
can be assumed, and HR(Z, B) = exp(b_1_ + b_4_B) is obtained. The HR varies with B due to its interaction with X, and the treatment effect is now a function of B rather than a constant.

The way that an exponential regression model accounts for treatment and covariate effects is similar to that of a Cox regression model, which is defined in terms of the well-known hazard function h(t) = h_0_(t) · exp(LIN), with no parametric form specified for the baseline hazard h_0_(t) and no intercept term included in LIN. One may think of h_0_(t) in the Cox model as a more general, time-varying form of e^b0^ in an exponential regression model. The exponential and Cox models thus have the same form for HR(Z, B), and both provide a basis for estimating hazards that vary with (X, Z, B) to use for optimizing each individual patient’s treatment. If, for example, a published paper reports a single control arm 12-month DFS probability for all patients, such as 30%, this implies that a Cox or exponential model with only b_1_X in the linear term has been assumed since otherwise, the estimated 12-month probabilities would vary with Z and B.

Risk prediction nomograms, such as the Assure nomogram, typically provide point estimates of probabilities such as P(Y > t | X = 0, Z = z, B = b). Table 3 shows that our simple treatment effect calculator, under the exponential model, closely predicts, with less than 5% error, the point estimates for the milestone survival probabilities with adjuvant treatment in recent phase 3 RCTs of adjuvant therapies, including KEYNOTE-564 [9,120,121,122,123]. Although we have used DFS in our examples, the treatment effect calculator can be used for any time-to-event endpoint, including DFS, OS, recurrence-free survival, and progression-free survival time. The Assure nomogram does not provide uncertainty quantification for its estimates of survival probabilities. It would be more desirable for such uncertainty in estimating P(Y > t | X, Z, B) to also be incorporated into the individualized treatment decision making procedure that we propose below.

## 5. Clinical Scenarios

In this section, we will illustrate how to use selection diagrams (Figure 8) and potential outcomes in Equation (3) to make adjuvant treatment decisions in RCC patients with particular covariates. This will be performed using the treatment effect calculator (Appendix A), statistical estimators computed from the KEYNOTE-564 RCT and the Assure RCC prognostic nomogram, and contextual causal knowledge of RCC biology [9,11,12,13,14,15,16,17,18,19]. While our treatment effect calculator relies on a very simple assumed model, in other clinical settings similar decisions may be made using more complex statistical regression models, provided that appropriate computer software is available [45,118,119]. This is because the selection diagrams, the do-calculus, and potential outcomes equations encode structural causal constraints, but they do not depend on particular assumptions underlying the statistical regression models used to provide patient-specific estimates [25,26,45].

In the illustrative examples, we will use DFS time as the clinical outcome because this was the primary endpoint of KEYNOTE-564 [9] and because surveys of RCC patients suggest that they value DFS and OS equally in the adjuvant setting [124]. An important causal issue is that a frontline adjuvant treatment’s effect on OS time is confounded by mediating effects of a salvage therapy given later at disease recurrence [10]. This would impact the effects of frontline adjuvant treatment on OS time in KEYNOTE-564, as well as baseline prognosis computed from the Assure nomogram. To use OS time as the primary outcome, the approach illustrated below might be applied more generally by accounting for the joint effects on OS time of (frontline, salvage) therapy pairs as two-stage dynamic treatment regimes, rather than ignoring salvage therapy. This requires more complex statistical models and methods but can provide much greater insights [45,66,125].

For illustration, we will assume that an absolute risk reduction (ARR) ≥ 5% in 24-month DFS probability is considered to be sufficient to offset the risk of toxicity and financial costs of adjuvant pembrolizumab versus surveillance. The 24-month cutoff corresponds to the published median follow-up time of 24.1 months in KEYNOTE-564 patients [9]. All patients described below are assumed to be otherwise healthy with no history of autoimmune diseases that would increase the risk of immune-related AEs due to adjuvant pembrolizumab. While many prognostic algorithms and nomograms have been developed to predict RCC outcomes following nephrectomy [11,126,127,128,129,130,131,132,133,134], in the examples below, we will use the Assure RCC prognostic nomogram due to its more contemporary validation and use of prospective data [11]. In the future, the accuracy of prognostic nomograms may be improved by incorporating observational and experimental knowledge from molecular biomarkers, such as circulating tumor DNA and cell-free methylated DNA [10,135,136].

### 5.1. Patient I

Our first example is a patient who would have been eligible for enrollment in KEYNOTE-564. Patient I is a 48-year-old man who underwent nephrectomy for a 10.6-cm clear cell RCC, pT3a, Fuhrman nuclear grade 4 with sarcomatoid features and coagulative necrosis, invading the renal vein with no pathological lymph node involvement or distant metastatic disease. These features would place Patient I in the intermediate-high risk category using the KEYNOTE-564 criteria shown in Table 1. Using the Assure nomogram, his estimated two-year DFS probability is 41.4%. There is no biological reason to assume effect modification of the HR estimate of KEYNOTE-564 for this patient. The statistical model used for the primary endpoint analysis of KEYNOTE-564 did not include interaction terms between any patient covariates and treatment [9]. It thus is assumed implicitly that the effect of treatment on the HR scale is stable across all patients enrolled with no differences due to a mediating effect of the RCC immune microenvironment. Therefore, Figure 8B can be reduced to the additive risk model shown in Figure 6B. Assuming an exponential or Cox regression model with LIN = log_e_(h) = b_0_ + b_1_·X + b_2_·Z implies that log_e_{h(0, Z)} = b_0_ + b_2_·Z when X = 0 (adjuvant pembrolizumab is not administered) and log_e_{h(1, Z)} = b_0_ + b_1_ + b_2_·Z when X = 1 (adjuvant pembrolizumab is administered), so HR = e^b1^ regardless of Z. Using the Assure nomogram estimate of 41.4% for the DFS probability at 24 months with surveillance (X = 0), since P(Y > 24 | X = 0, Z) = exp{−24·h(0, Z) }, this corresponds to the estimated hazard h(0, Z) = − log_e_(0.414)/24 = 0.03675. We next use the treatment effect calculator (Appendix A) to obtain an estimate of the hazard h(1, Z) of the patient if he were treated with adjuvant pembrolizumab, based on the estimated pembrolizumab-versus-surveillance HR = 0.680 obtained from KEYNOTE-564. Since HR = h(1, Z)/h(0, Z), the estimated hazard with adjuvant pembrolizumab is h(1, Z) = HR h(0, Z) = 0.680·0.03675 = 0.02499, which gives estimated 24-month DFS probability e^−(0.02499·24)^ = 0.549, or 54.9%. These computations give estimated ARR = 54.9% − 41.4% = 13.5% in 24-month DFS probability if adjuvant pembrolizumab is administered (Table 4). Given this substantial, expected clinical benefit, adjuvant pembrolizumab would appear to be a very reasonable choice for Patient I. Note that these numerical computations mimic the theoretical derivation, given earlier, of an unbiased causal effect estimate based on a patient’s two potential outcomes when comparing treatments. An important general point is that being able to compute a 95% CI, or a 95% credible interval under a Bayesian model [58,64], around the estimated ARR to account for uncertainty would make it much more useful. For example, the estimate 13.5% with 95% CI [−2.0%, 28.0%] would be far less reliable than if the 95% CI were [10.0%, 16.0%]. This underscores the need for more extensive software that can account for variability of estimates.

The above computations require one to assume that the relative treatment effect should not differ between the patient and the population enrolled in KEYNOTE-564 due to a different RCC histology. Because Patient I has clear cell RCC histology and perfectly fulfills the enrollment criteria of KEYNOTE-564 [9], we can make this assumption regarding the stability of the HR, which allows us to input the estimated KEYNOTE-564 HR for DFS of 0.68 in our treatment effect calculator (Appendix A).

### 5.2. Patient II

Patient II is a 48-year-old woman who underwent nephrectomy for a 7.0-cm clear cell RCC, pT3a, Fuhrman nuclear grade 2 without sarcomatoid features or coagulative necrosis, invading the renal vein with no pathological lymph node involvement or distant metastatic disease. These features place Patient II in the intermediate-high risk category of KEYNOTE-564 (Table 1), the same risk category as Patient I. However, her predicted two-year DFS probability by the Assure nomogram is 87.2%, since the nomogram accounts for her favorable prognostic covariates. She perfectly fulfills the enrollment criteria of KEYNOTE-564 and again we can use the HR for DFS estimated by this phase 3 RCT using the additive effect model represented by Figure 6B. Her predicted 24-month DFS probability with adjuvant pembrolizumab is 91.1%, and following the same computational steps as given above for Patient I, this gives an ARR of only 3.9% (Table 4). Considering the risk of toxicity, adjuvant pembrolizumab is much less attractive for Patient II compared with Patient I, even though both patients would be classified as intermediate-high risk by KEYNOTE-564. This example underscores the importance of estimating the amount of potential benefit for each patient using their individual covariates.

### 5.3. Patient III

Patient III is a 55-year-old man whose nephrectomy revealed a 10.3-cm clear cell RCC, pT2b, Fuhrman nuclear grade 3 with coagulative necrosis but no sarcomatoid features, and no vascular invasion, pathological lymph node involvement, or metastatic disease. This patient would not be eligible for enrollment in KEYNOTE-564 because his disease would not be deemed sufficiently high risk using the criteria summarized in Table 1. However, the predicted 24-month DFS probability from the Assure nomogram is 68%, which is much worse than that of Patient II. This is because the eligibility criteria of clinical trials are not designed to have the discriminative performance of validated prognostic risk prediction models such as the Assure nomogram. Although Patient III would not be eligible for KEYNOTE-564, it is plausible that the treatment effect of adjuvant pembrolizumab on the HR scale would be similar to that of Patients I and II, who had the same RCC histology as Patient III, based on medical knowledge. Using the additive modeling framework of Figure 6B, the treatment effect calculator yields a predicted 24-month DFS probability of 76.9%, and ARR = 76.9% − 68% = 8.9% with adjuvant pembrolizumab (Table 4). Therefore, adjuvant pembrolizumab would be a reasonable choice for Patient III, certainly more so than for Patient II, despite the fact that Patient III would have been ineligible for enrollment in KEYNOTE-564. This example illustrates how personalized treatment decisions can be guided by combining causal, statistical, and medical thinking applicable to each patient, and not simply by considering clinical trial eligibility status.

### 5.4. Patient IV

Patient IV is a 52-year-old woman who underwent surgery to remove a solitary right upper lobe lung metastasis 6 months after nephrectomy for a 10.2-cm clear cell RCC, pT3a, Fuhrman nuclear grade 3 without sarcomatoid features or coagulative necrosis, invading the renal vein with no pathological lymph node involvement. She would have been eligible for KEYNOTE-564 per the criteria outlined in Table 1 given her M1 NED status following removal of the lung metastasis. Accordingly, the additive modeling framework represented in Figure 6B can be used. However, the statistical regression model underlying the Assure nomogram did not include M1 NED status as a covariate [11]. For example, patients with metachronous RCC metastasis occurring many years after the nephrectomy and successfully treated with surgery or radiation therapy may have a lower probability of subsequent recurrence than patients whose solitary metastasis occurred early after nephrectomy [137,138,139]. None of the currently available prognostic algorithms or nomograms can discriminate between such scenarios. Consequently, estimates of DFS probabilities for Patient IV that account for M1 NED status are not available [11,126,127,128,129,130,131,132,133,134]. In this situation, Figure 6 tells us what data are needed and how to develop statistical regression models using observations from large datasets that are representative of Patient IV (Table 4). What we currently lack to make inferences for this patient is not predictive biomarkers generated from interactive models, but rather simple prognostic risk models that can risk stratify different M1 NED scenarios. Large multi-institutional observational datasets of the recurrence risk of M1 NED patients not treated with any adjuvant therapy would suffice to develop such M1 NED nomograms.

### 5.5. Patient V

Patient V is a 54-year-old woman who underwent nephrectomy for a 13.9-cm papillary type I RCC, pT3a, Fuhrman nuclear grade 2, without sarcomatoid features or coagulative necrosis, invading the renal vein with no pathological lymph node involvement or distant metastatic disease. Other than the different histology, all other features would render this patient eligible for KEYNOTE-564 in the intermediate-high risk group (Table 1). Given that the FDA approval of adjuvant pembrolizumab was not restricted to a specific RCC histology, a clinician might be tempted to recommend adjuvant pembrolizumab for Patient V. However, the estimated 24-month DFS probability from the Assure nomogram for Patient V is 94.6% with surveillance. Papillary type I RCC is a histology with a different immune microenvironment that is less likely than that of clear cell RCC to produce responses to immune checkpoint therapies such as pembrolizumab [8,16,140]. Therefore, for this patient, Figure 8B cannot be reduced to Figure 6B. Even if we assume the unlikely best case scenario where the effect of adjuvant pembrolizumab is even better in papillary type I RCC than in clear cell RCC, corresponding for example to HR = 0.5 for pembrolizumab versus placebo, the predicted 24-month DFS probability with adjuvant pembrolizumab would be 97.3%, corresponding to ARR = 2.7%. Thus, it is very unlikely that the benefit of adjuvant pembrolizumab would be clinically meaningful for this patient.

### 5.6. Patient VI

Patient VI is a 49-year-old man whose nephrectomy revealed a 10.4-cm papillary type II RCC, pT3a, Fuhrman nuclear grade 4, with coagulative necrosis and sarcomatoid features, invading the renal vein with no pathological lymph node involvement or distant metastatic disease. Similar to the papillary type I RCC found in Patient V, papillary type II RCC harbors a distinct immune microenvironment that is less likely than that of clear cell RCC to respond to pembrolizumab [8,16,140]. Because KEYNOTE-564 did not enroll patients with papillary type II RCC, Figure 8B instructs us to obtain experimental data on the effect of adjuvant pembrolizumab in the immune microenvironment of papillary type II RCC. Clinical trial results of adjuvant pembrolizumab in this setting are not available. However, it is known that the efficacy of immune checkpoint therapies such as pembrolizumab is reduced by ~50% in patients with metastatic papillary type II RCC compared with metastatic clear cell RCC [8]. Using the Assure nomogram, the predicted 24-month DFS probability of Patient VI is only 41.4%. Even if we assume that the HR for DFS is only 0.84 [i.e., equal to 0.68 + (1–0.68)/2] in patients with papillary type II RCC, the predicted 24-month DFS probability with adjuvant pembrolizumab for Patient VI would be 47.7%, corresponding to an ARR of 6.3%. Thus, adjuvant pembrolizumab would not be an unreasonable option for this patient. If desired, this computation could be repeated for each of a set of hypothetical HR values, such as 0.68, 0.72, 0.76, 0.80, and 0.84, as a sensitivity analysis using our treatment effect calculator (Appendix A). By making additional assumptions here, we have approximated the potential effect modification due to the interaction between adjuvant pembrolizumab and RCC histology in papillary type II RCC, despite the lack of experimental studies of adjuvant pembrolizumab in papillary type II RCC.

### 5.7. Patient VII

Patient VII is a 56-year-old woman who underwent nephrectomy for a 9.5-cm chromophobe RCC, pT2a, without coagulative necrosis, sarcomatoid features, vascular invasion, pathological lymph node involvement, or distant metastatic disease. Due to its inherent nuclear atypia, chromophobe RCC is not normally assigned a Fuhrman nuclear grade, but the lack of sarcomatoid features or necrosis is compatible with a low-grade chromophobe RCC [141,142]. A low nuclear grade thus can be selected in the Assure nomogram. KEYNOTE-564 did not enroll patients with chromophobe RCC, which has a very different immune microenvironment from clear cell RCC and typically responds very poorly to immune checkpoint therapies such as pembrolizumab [8,16,140]. Figure 8B tells us that we need experimental data to properly estimate the treatment effect of adjuvant pembrolizumab on the distinct immune microenvironment of chromophobe RCC. No such data are currently available. However, the predicted 24-month DFS probability from the Assure nomogram for Patient VII is 97.9%. Even if we assume the unlikely scenario whereby the treatment effect of adjuvant pembrolizumab for chromophobe RCC is even higher than that for clear cell RCC, corresponding for example to HR = 0.5 for pembrolizumab versus placebo, the predicted 24-month DFS probability with adjuvant pembrolizumab would be 98.9% corresponding to an ARR of only 1.0%. Thus, adjuvant pembrolizumab is not an appropriate option for Patient VII under any treatment effect scenario.

### 5.8. Patient VIII

The final patient is a 52-year-old man whose nephrectomy revealed an 11.4-cm chromophobe RCC, pT2b, with sarcomatoid features but without coagulative necrosis, vascular invasion, pathological lymph node involvement, or distant metastatic disease. The presence of sarcomatoid features is compatible with a high-grade chromophobe RCC [141,142], and thus a high nuclear grade can be selected in the Assure nomogram. As with Patient VII, we are instructed by Figure 8B that we lack the necessary experimental knowledge to properly estimate the treatment effect of adjuvant pembrolizumab for a patient with chromophobe RCC. Furthermore, the 24-month DFS probability estimated by the Assure nomogram is 91.3%, which is unusually favorable for chromophobe RCC with sarcomatoid features. Additional observational studies focusing on chromophobe RCC suggest that the rare situations where sarcomatoid features are present, as in Patient VIII, can decrease 24-month DFS probability to less than 50%, particularly in a pT2b tumor [143]. The Assure dataset did not include enough cases of chromophobe RCC with sarcomatoid features to discriminate between such scenarios [11]. This serves as an example of how a structured causal framework can facilitate model checking. The Assure nomogram yielded an implausible DFS probability, pointing to the need for additional observational studies to generate prognostic risk scores capable of discriminating between rare attributes of chromophobe RCC, such as the presence of sarcomatoid features.

## 6. Conclusions

We have described how structural causal considerations can inform both patient-centered inferences and study design and illustrated how this may be performed with several practical clinical examples. As summarized in Table 4, this framework allowed us to integrate the complementary features of the KEYNOTE-564 phase 3 RCT, observational data from the Assure prognostic risk nomogram, and corollary biological knowledge on the immune microenvironment of different RCC histologies. Using causal diagrams and the corresponding do-calculus syntax, we were able to decide whether the transportability of knowledge from different domains to each patient scenario was plausible. In cases where complete domain knowledge was not available to allow unbiased estimation of the treatment effect for a specific patient, we were able to identify the pertinent knowledge gaps and the types of studies, observational or experimental, that would be needed to license the desired transport. The generation of patient-specific estimates in each scenario was based on the potential outcomes framework expressed in Equations (1) and (3), along with statistical regression modeling assumptions. In Figure 8B, we assumed that the only mediator for the treatment effect of adjuvant pembrolizumab is the RCC immune microenvironment. As our knowledge of the relevant biological events evolves, the causal chain may become more complex, possibly including new mediators and biomarkers. In such a case, the choice of whether the S-admissibility criterion is fulfilled for adjuvant RCC will again depend on the causal context in which population differences are embedded [35,36,37].

Clear cell is the most common RCC histology, representing approximately 75% of all cases. Using available knowledge to determine and represent plausible causal relationships allowed us to identify clear cell RCC patients for whom adjuvant pembrolizumab would be warranted despite their not fulfilling the eligibility criteria of the KEYNOTE-564 trial (Patient III) or, conversely, would not be warranted despite their classification as intermediate-high risk by KEYNOTE-564 (Patient II). Since FDA approval of adjuvant pembrolizumab was not limited to clear cell RCC, clinicians also will be tasked with deciding whether to recommend adjuvant pembrolizumab in other RCC histologies. These include papillary RCC (approximately 10%–20% of cases) and chromophobe RCC (approximately 5% of cases), the second and third most common RCCs, respectively [8].

Practicing clinicians must regularly make treatment decisions using each patient’s covariates along with information from diverse sources, including clinical trials, laboratory experiments, observational data, and biological knowledge. Causal diagrams are intuitive tools that can be used by clinicians to incorporate such domain knowledge to inform their decisions. Using the selection diagram in Figure 8B, we exploited established knowledge of biological commonalities and differences between RCC histologies learned from designed experiments and observational data, to provide recommendations for a variety of different RCC patients. These included patients with papillary RCC (patients V and VI) and chromophobe RCC (patient VII).

The proposed framework describes the causal assumptions necessary to utilize the additive or interactive effect modeling approaches recommended by the PATH statement to generate patient-specific predictions of treatment effects obtained from RCTs [67,68]. The general framework is not restricted to RCC and can be applied in a wide variety of different medical contexts, as intended by PATH [67,68]. While the interactive effect modeling approach used by PATH requires the use of statistical treatment–subgroup interactions, our causal framework can accommodate more general scenarios, such as making treatment decisions involving HER2-targeted agents for breast cancer (Figure 7A) based on preclinical research.

## Figures and Tables

**Figure 1 cancers-14-03923-f001:**
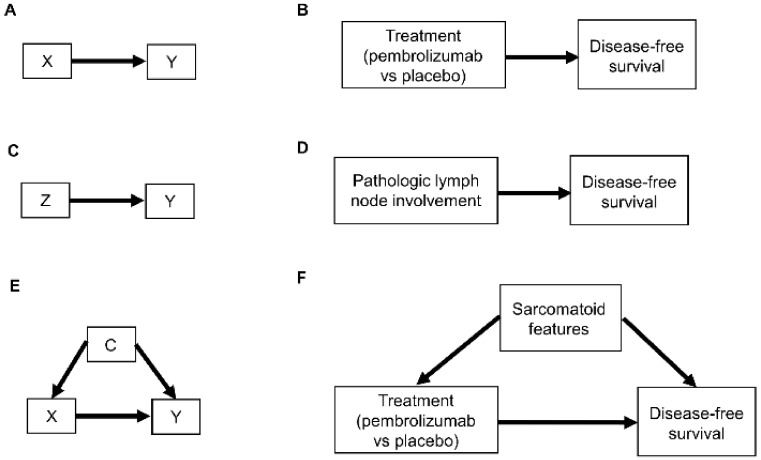
Examples of causal relationships represented by DAGs: (**A**) A simple scenario whereby the treatment choice X (also known as “exposure”) directly influences the outcome of interest Y; (**B**) Corresponding simple clinical scenario whereby the treatment choice between adjuvant pembrolizumab or placebo directly influences the outcome of disease recurrence or death measured by DFS; (**C**) Simple scenario whereby a prognostic variable Z directly influences the outcome Y; (**D**) Corresponding clinical scenario whereby the presence or absence of pathologic lymph node involvement directly influences DFS; (**E**) Scenario whereby a confounder C directly influences both the treatment choice X and the outcome Y; (**F**) Corresponding clinical scenario whereby the presence or absence of sarcomatoid features acts as a confounder by influencing treatment choice and DFS.

**Figure 2 cancers-14-03923-f002:**
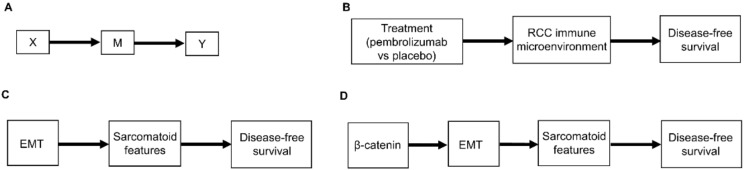
(**A**) Scenario whereby the effect of the treatment choice X on the outcome Y is transmitted by a mediator M; (**B**) Corresponding clinical scenario whereby the effect of treatment choice on DFS is transmitted by the renal cell carcinoma (RCC) immune microenvironment; (**C**) Another clinical scenario whereby the effect of epithelial-mesenchymal transition (EMT) on DFS is transmitted by the presence or absence of sarcomatoid features; (**D**) More detailed version whereby the effect of β-catenin levels on DFS is mediated by EMT and the resultant presence or absence of sarcomatoid features.

**Figure 3 cancers-14-03923-f003:**
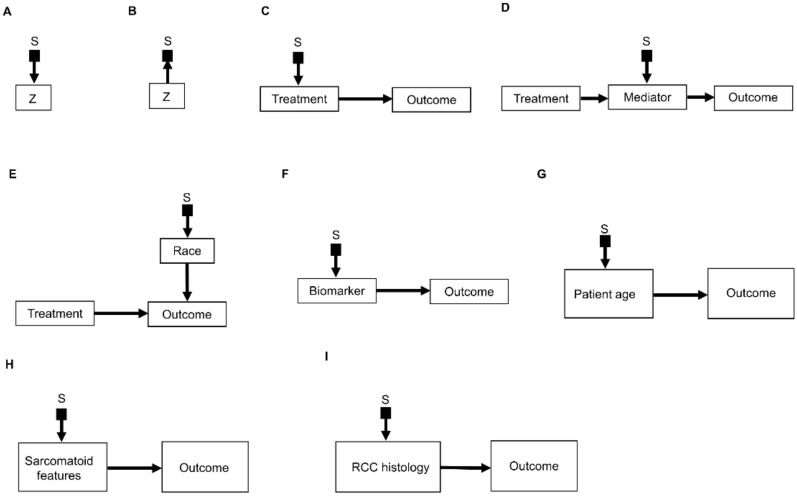
Examples of causal relationships represented by selection diagrams to guide transportability of causal effects across populations: (**A**) Selection node S indicating shifts across populations in the variable Z. In this scenario, the possible values of Z inherently differ between populations; (**B**) Selection node S indicating selection bias for the variable Z. In this scenario, the differences in Z between populations are due to sampling biases and not due to inherent variation; (**C**) Selection node S indicating treatment shifts across populations; (**D**) Selection node S indicating mediator shifts across populations; (**E**) Selection node S indicating shifts in the race of patients treated in a randomized controlled trial (RCT) performed in the United States, compared with an RCT performed in China. In this diagram, the variable “race” directly influences the outcome of interest and the race of patients enrolled is different between the two RCTs; (**F**) Selection node S indicating shifts in biomarker values across populations; (**G**) Selection node S indicating shifts in patient age across populations; (**H**) Selection node S indicating shifts in the presence or absence of sarcomatoid features across populations; (**I**) Selection node S indicating shifts in RCC histology across populations.

**Figure 4 cancers-14-03923-f004:**
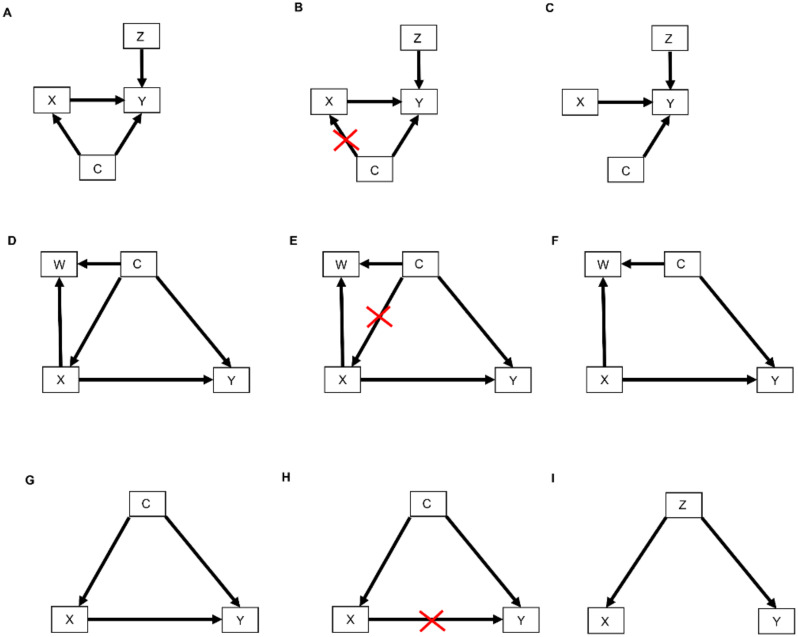
DAGs modified by the do() operator: (**A**) Treatment choice X influences the outcome Y which is also influenced by covariates Z. Furthermore, unknown confounders C may influence both treatment choice X and the outcome Y; (**B**) As indicated by the red crossmarks, the do() operator expression do(X = x) removes all arrows entering X. This modification represents the physical effect of an experimental intervention that sets the value X = x as constant while keeping the rest of the causal model unchanged; (**C**) This yields a new DAG whereby the distribution P(Y | do(X = x), Z) is the same as P(Y | X = x, Z); (**D**) In this DAG, treatment choice X influences the outcome Y and another covariate W. The confounder C influences X, Y, and W; (**E**) The do() operator expression do(X = x) removes all arrows entering X; (**F**) This yields a new DAG whereby the distribution P(Y | do(X = x), C) is the same as P(Y | X = x, C); (**G**) The confounder C influences the treatment choice X and the outcome Y; (**H**) Paths from X to Y that remain after removing all arrows pointing out of X are called “backdoor” paths because they flow backward out of X into Y. Shown here is the backdoor path from X to Y via the confounder C; (**I**) In this DAG, there are no forward-directed arrows pointing out of X toward Y. Therefore, according to rule 3 of the do-calculus, P(Y | do(X = x, Z) = P(Y | Z).

**Figure 5 cancers-14-03923-f005:**
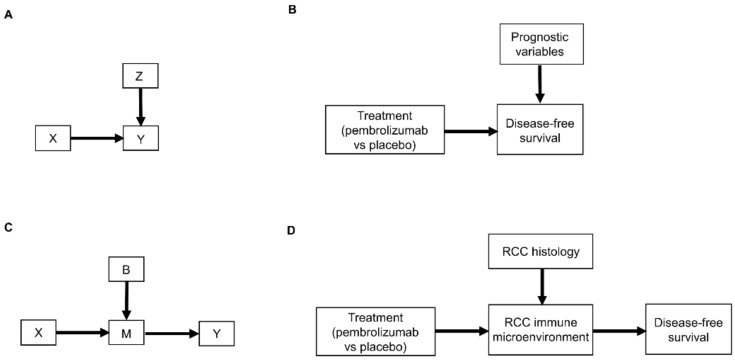
DAGs representing the two modeling strategies recommended by the Predictive Approaches to Treatment effect Heterogeneity (PATH) statement: (**A**) Additive (“risk modeling”) approach where the outcome Y is independently influenced by a treatment X and baseline risk Z; (**B**) Additive modeling scenario whereby DFS is independently influenced by the treatment choice between adjuvant pembrolizumab or placebo and by each patient’s baseline risk as determined by renal cell carcinoma (RCC) prognostic risk models such as the Assure nomogram; (**C**) Interactive “effect modeling” approach where treatment X and baseline biomarker B interact by influencing together the mediator M, transmitting the effect of X on the outcome Y; (**D**) Interactive modeling scenario whereby RCC histology interacts with the treatment choice between adjuvant pembrolizumab or placebo by influencing together the RCC immune microenvironment transmitting the effect of treatment choice on DFS.

**Figure 6 cancers-14-03923-f006:**
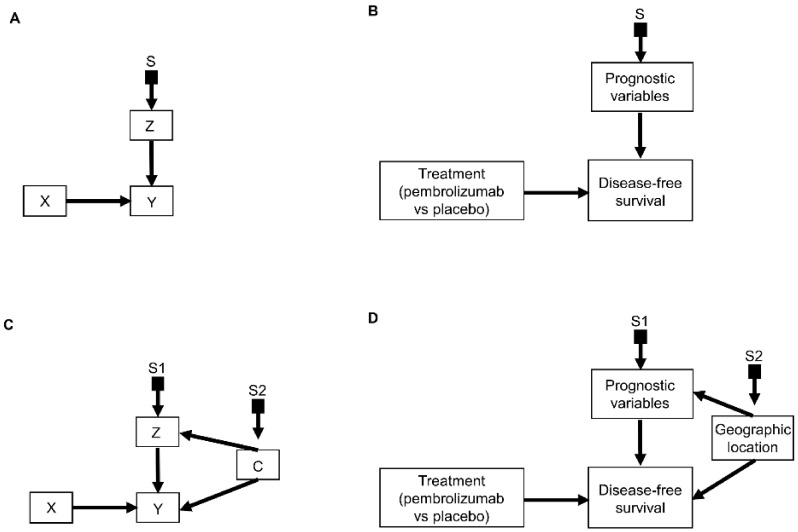
Selection diagrams representing the risk modeling approach recommended by the Predictive Approaches to Treatment effect Heterogeneity (PATH) statement: (**A**) Risk modeling approach where the outcome Y is determined by a randomized treatment X and baseline risk Z. The selection node S indicates that baseline risk can shift across populations; (**B**) Risk modeling scenario whereby DFS is influenced by the randomized treatment choice between adjuvant pembrolizumab or placebo and by each patient’s baseline risk as determined by RCC prognostic risk models such as the Assure nomogram. The selection node S indicates that baseline prognostic variables for DFS risk can shift across populations; (**C**) Scenario where one or more confounders C may influence baseline risk Z and treatment outcome Y. The selection nodes S1 and S2 indicate that baseline risk Z and confounder C, respectively, can shift across populations; (**D**) Corresponding clinical scenario where geographic location influences both the baseline prognostic risk and the survival outcomes of patients with RCC. The selection nodes S1 and S2, respectively, indicate that baseline prognostic variables and geographic location can shift across populations.

**Figure 7 cancers-14-03923-f007:**
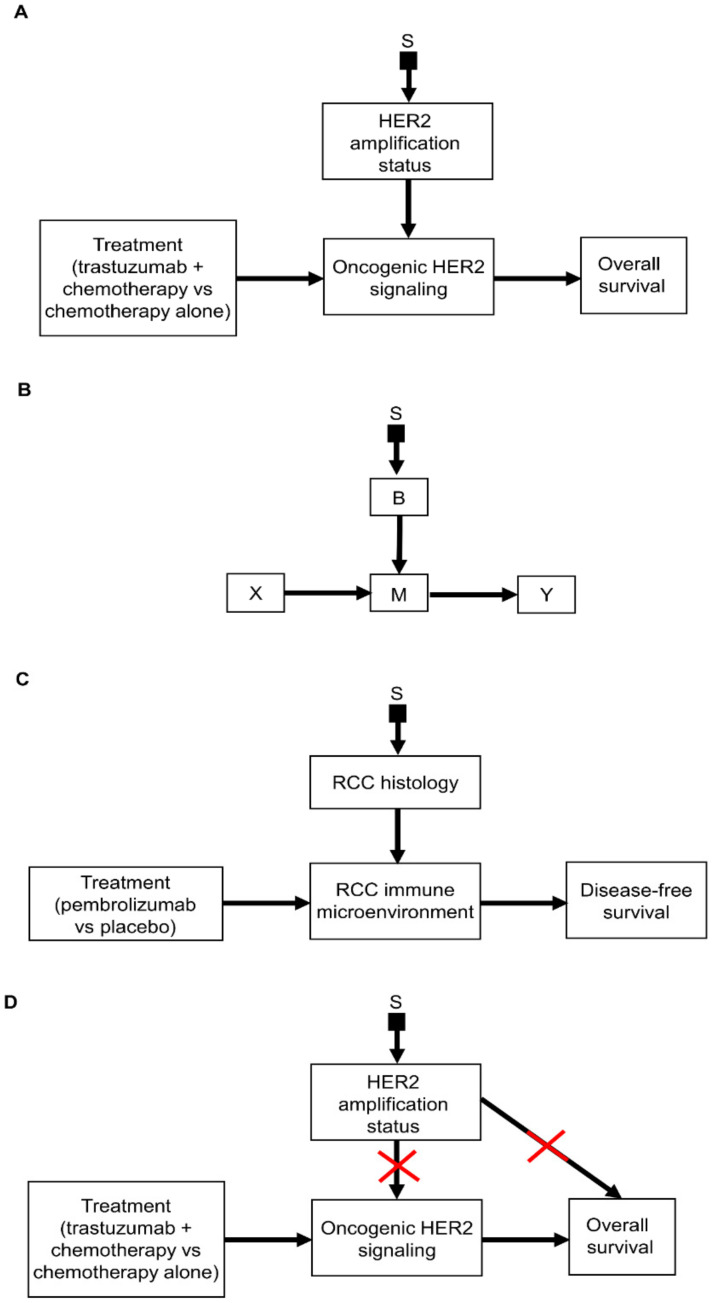
Selection diagrams representing biological treatment effect modification: (**A**) Clinical scenario of a randomized clinical trial comparing the survival outcomes of trastuzumab, an inhibitor of human epidermal growth factor receptor 2 (HER2) signaling, in combination with chemotherapy versus chemotherapy alone in patients with metastatic breast cancer. *HER2* amplification status influences oncogenic HER2 signaling, mediating the treatment effect. The selection node S denotes that *HER2* amplification status can vary across populations; (**B**) Biological treatment effect modification framework where the effect of a randomized treatment X on the outcome Y is mediated by M. The mediator M is influenced by the biomarker B, which can vary across populations, as denoted by the selection node S; (**C**) Clinical scenario whereby the effect of the randomized treatment choice between adjuvant pembrolizumab or placebo on DFS is mediated by the RCC immune microenvironment, which is influenced by RCC histology. The latter can vary across populations, as denoted by the selection node S; (**D**) Clinical scenario where *HER2* amplification status is both a predictive biomarker (as shown by the arrow toward oncogenic HER2 signaling) and a prognostic biomarker (as shown by the direct arrow toward overall survival). Covariate weighting for *HER2* amplification status will block both pathways (as indicated by the red crossmarks) from the selection node S to the overall survival outcome.

**Figure 8 cancers-14-03923-f008:**
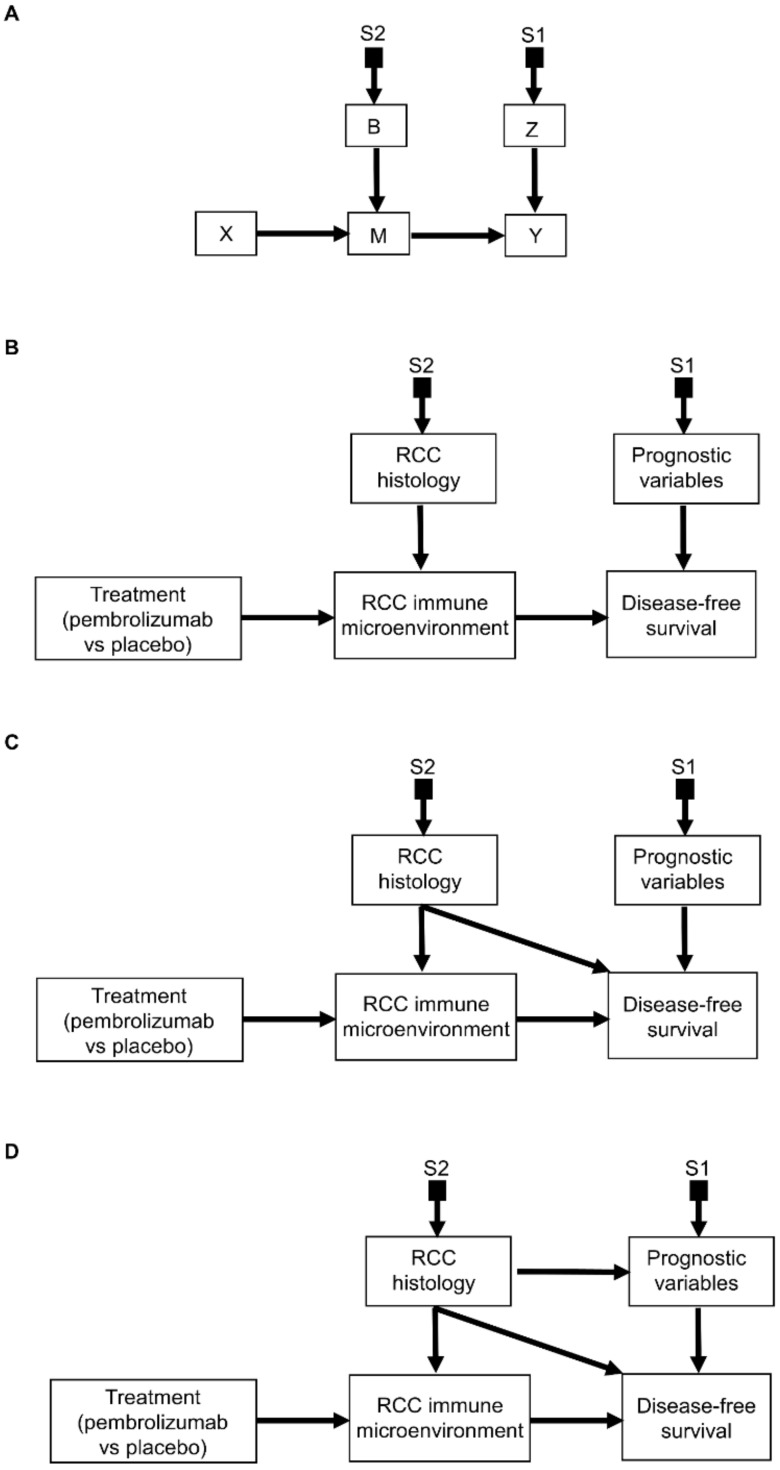
Selection diagrams representing a general framework for treatment effect heterogeneity modeling: (**A**) Selection diagram combining both risk modeling and biological treatment effect modification. Patient populations may differ in baseline risk, denoted by the selection node S1, and/or biomarkers B that affect biological mediators M of the treatment effect, as denoted by the selection node S2; (**B**) Corresponding clinical scenario where risk and effect modeling may be used alone or in combination depending on the characteristics of each patient seen in the clinic. Patient populations may differ in baseline DFS risk, denoted by the selection node S1, and/or RCC histology, as denoted by the selection node S2. The S-admissibility criterion for this scenario remains the same even in the more complex scenario whereby the RCC histology also directly influences DFS as shown in (**C**) and the prognostic variables as shown in (**D**).

**Figure 9 cancers-14-03923-f009:**
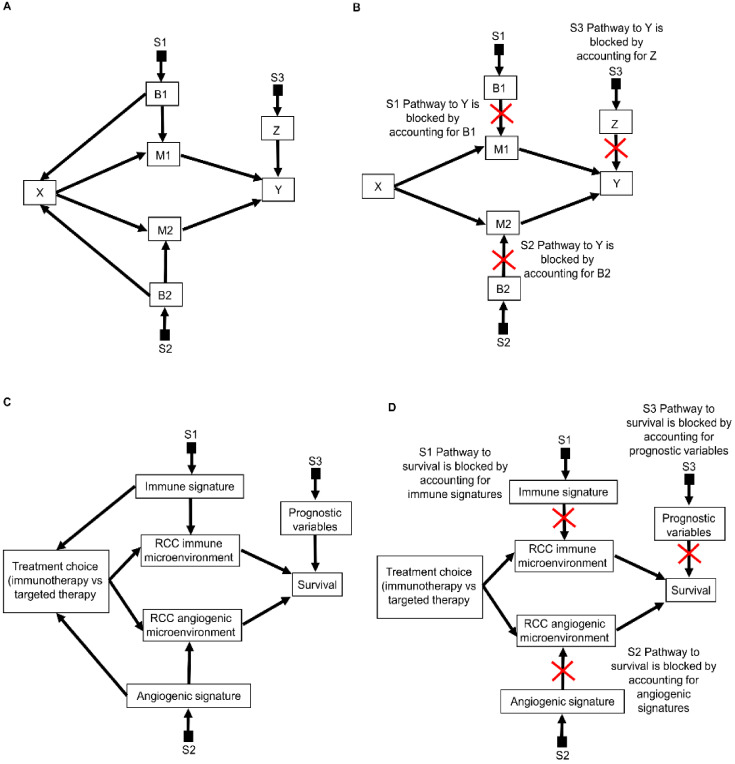
Selection diagrams representing more complex scenarios of treatment effect heterogeneity: (**A**) The effect of treatment choice X on the outcome Y is mediated by two mediators, M1 and M2, which are influenced by biomarkers B1 and B2, respectively. Prognostic variables Z also independently influence the outcome Y. B1 and B2 also influence the treatment choice X. Patient populations may differ in B1 (denoted by S1), B2 (denoted by S2) and in Z (denoted by S3); (**B**) The same selection diagram modified to illustrate the considerations behind the S-admissibility criterion licensing the transportation of knowledge toward the population of patients seen in the clinic. S-admissibility states that once all arrows pointing to the treatment variable X are removed, we need to covariate weight for all variables needed to block all causal pathways that transmit effects from each selection node S toward the outcome Y. In this example, covariate weighting for B1, B2, and Z, respectively, blocks the pathways (as indicated by the red crossmarks) from S1 to Y, S2 to Y, and S3 to Y; (**C**) Corresponding clinical scenario where patients with metastatic RCC may be treated with either immunotherapy or anti-angiogenic targeted therapy. The effect of choosing between immunotherapy or targeted therapy is mediated by the RCC immune microenvironment, influenced by a baseline biological immune signature, which shifts across studied populations as denoted by S1. The effect of choosing between the two treatments is also mediated by the RCC angiogenic microenvironment, influenced by a baseline angiogenic signature, which shifts across studied populations as denoted by S2. The two signatures also influence the treatment choice in the studied populations. The survival outcome is also independently influenced by prognostic variables which shift across populations as denoted by S3; (**D**) The S-admissibility criterion licenses the transportation of knowledge toward the population of patients seen in the clinic if we properly covariate weight for shifts in prognostic variables and in immune and angiogenic signatures.

**Table 1 cancers-14-03923-t001:** Prespecified disease risk categories for each enrolled patient used by the KEYNOTE-564 randomized clinical trial of adjuvant pembrolizumab versus placebo control in patients with clear cell renal cell carcinoma [9].

	Intermediate-High Risk		High Risk		M1 with No Evidence of Disease
Pathologic primary tumor (T) stage	pT2	pT3	pT4	Any pT	Any pT
Tumor nuclear grade	Grade 4 or sarcomatoid	Any grade	Any grade	Any grade	Any grade
Regional lymph node (N) stage	N0	N0	N0	N1	Any lymph node stage
Metastatic stage	M0	M0	M0	M0	M1
pT2: primary tumor >7 cm in greatest dimension, limited to the kidney pT3: primary tumor extends into major veins or perinephric tissues, but not into the ipsilateral adrenal gland and not beyond Gerota’s fascia pT4: Tumor invades beyond Gerota’s fascia (including contiguous extension into the ipsilateral adrenal gland) N0: No regional lymph node metastasis N1: Metastasis in regional lymph node(s) M0: No history of radiologically visible distant metastasis M1: History of radiologically visible distant metastasis					

**Table 2 cancers-14-03923-t002:** The ladder of causation [40]. Y is the outcome, DFS time; X is treatment, with X = 1 for adjuvant pembrolizumab and X = 0 for placebo; Z is the baseline prognostic risk, with Z = 1 for high risk of recurrence. ccRCC = clear cell renal carcinoma.

Layer	Activity	Analysis Unit	Mathematical Expression	Example Query
One	Observation	Population	P(Y | Z)	What is the DFS time distribution in patients at high risk for ccRCC recurrence?
Two	Intervention	Population	P(Y | do(X = 1))	What is the DFS time distribution in patients with ccRCC treated with adjuvant pembrolizumab?
Three	Potential outcomes	Individual Patient	E(Y^X = 1^ | Z = 1) − E(Y^X = 0^ | Z = 1)	What would the expected DFS time be if I treat a patient with high-risk ccRCC with adjuvant pembrolizumab compared to placebo?

**Table 3 cancers-14-03923-t003:** Estimated and reported milestone survival probabilities using the treatment effect calculator (Appendix A) assuming an exponential survival distribution in adjuvant therapy trials. DFS = disease-free survival, HR = hazard ratio.

Type of Adjuvant Therapy	Milestone Time (Months)	Reported Milestone DFS Probability in Control Group	Estimated HR and CIs for DFS	Estimated Milestone DFS Probability in Treatment Group	Reported Milestone DFS Probability in Treatment Group	Difference between Estimated Versus Reported Milestone DFS Probability	Reference
Immune checkpoint therapy vs. placebo	12	76.2%	0.68	83.1%	85.7%	−2.6%	[9]
Immune checkpoint therapy vs. placebo	24	68.1%	0.68	77%	77.3%	−0.3%	[9]
Immune checkpoint therapy vs. placebo	6	60.3%	0.70	70.2%	74.9%	−4.7%	[120]
Chemotherapy vs. placebo	36	46%	0.45	70.5%	71%	−0.5%	[121]
Targeted therapy vs. placebo	24	52%	0.20	87.7%	89%	−1.3%	[122]
Targeted therapy vs. placebo	36	80.4%	0.57	88.3%	87.5%	+0.8%	[123]

**Table 4 cancers-14-03923-t004:** Patient-specific inferences and research design recommendations in eight clinical scenarios where adjuvant pembrolizumab is considered, based on KEYNOTE-564 (KN-564) [9] in patients harboring one of the three most common renal cell carcinomas (RCC). All patients are otherwise healthy and are interested in an absolute risk reduction (ARR) of at least 5% in two-year DFS probability with adjuvant pembrolizumab as estimated using the Assure prognostic nomogram [11].

Patient	RCC Histology	Eligible for KN-564	Age	Tumor Stage	Tumor Size (cm)	Fuhrman Nuclear grade	Necrosis	Renal Vein Invasion	Sarcomatoid features	Predicted 2-Year DFS with Surveillance	Predicted 2-Year DFS with Pembrolizumab	ARR	Recommend Adjuvant Pembrolizumab	External Observational Studies Needed	External Experimental Studies Needed
I	Clear cell	Yes	48	pT3a pN0 M0	10.6	4	Yes	Yes	Yes	41.1%	54.9%	13.5%	Yes	No	No
II	Clear cell	Yes	48	pT3a pN0 M0	7.0	2	No	Yes	No	87.2%	91.1%	3.9%	No	No	No
III	Clear cell	No	55	pT2b pN0 M0	10.3	3	Yes	No	No	68%	76.9%	8.9%	Yes	No	No
IV	Clear cell	Yes	52	pT3a pN0 M1 NED	10.2	3	No	Yes	No	Not estimable	Not estimable	Not estimable	Not estimable	Yes	No
V	Papillary type I	No	54	pT3a pN0 M0	13.9	2	No	Yes	No	94.6%	Not formally estimable but would be 97.3% even if HR = 0.5	Not formally estimable but would be 2.7% even if HR = 0.5	No	No	Yes
VI	Papillary type II	No	49	pT3 pN0 M0	10.4	4	Yes	Yes	Yes	41.4%	Not formally estimable but would be 47.7% even if hazard ratio (HR) = 0.84	Not formally estimable but would be 6.3% even if HR = 0.84	Not formally estimable but is a plausible recommendation under current state of knowledge	No	Yes
VII	Chromo-phobe	No	56	pT2a pN0 M0	9.5	Low grade	No	No	No	97.9%	Not formally estimable but would be 98.9% even if HR = 0.5	Not formally estimable but would be 1% even if HR = 0.5	No	No	Yes
VIII	Chromo-phobe	No	52	pT2b pN0 M0	11.4	High grade	No	No	Yes	Not estimable	Not estimable	Not estimable	Not estimable	Yes	Yes
pT2a: primary tumor >7 cm but ≤10cm in greatest dimension, limited to the kidney pT2b: primary tumor >10 cm in greatest dimension, limited to the kidney pT3a: primary tumor extends into the renal vein or its segmental (muscle containing) branches, or tumor invades perirenal and/or renal sinus fat (ie, perinephric fat), but not into the ipsilateral adrenal gland and not beyond Gerota’s fascia pN0: No regional lymph node metastasis M0: No history of radiologically visible distant metastasis M1 NED: History of radiologically visible distant metastasis with currently no evidence of disease															

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
