# Peer review of "A Causal Framework for Making Individualized Treatment Decisions in Oncology"

_cancers, 2022, doi:10.3390/cancers14163923_

Round 1
Reviewer 1 Report
Outstanding comprehensive review of the utility of causal diagrams for deciding whether experimental or observational data should be combined with RCT results to guide individual clinical decision making with real world illustrative cases. This is a real contribution summarizing the growing literature on this topic. One possible suggestion is a somewhat more detailed Summary (1-2 pages) for those without time to go through the entire paper. This is merely a suggestion to put to the authors
Author Response
Comment: “Outstanding comprehensive review of the utility of causal diagrams for deciding whether experimental or observational data should be combined with RCT results to guide individual clinical decision making with real world illustrative cases. This is a real contribution summarizing the growing literature on this topic. One possible suggestion is a somewhat more detailed Summary (1-2 pages) for those without time to go through the entire paper. This is merely a suggestion to put to the authors “
Author response: We thank the reviewer for their feedback. We have accordingly now substantially expanded the first-page Summary.
Reviewer 2 Report
Overall excellent methodologic paper that rationally describes how many providers incorporate patient data in clinical decision making. The authors first broadly describe the fundamentals of causal diagrams and then demonstrate how these principles can be used specifically for decision making in adjuvant pembro for patients with RCC. Overall the paper is detailed and well written. Dissemination of this thinking and approach is important to advance the field and better care for our patients. No major criticisms for the paper.
From an implementation standpoint, if it is possible to convert the excel sheet into an online calculator it would be easier for providers to make more individualized decisions for patients. Ideally, it could be incorporated into the FoxChase (https://cancernomograms.com/nomograms/) nomogram site where the ASSURE RCC calculator also lives (as listed in the introduction). That would could allow for seamless integration of available data from ASSURE RCC into the treatment effect calculator.
Otherwise, the authors are to be congratulated on an excellent work.
Author Response
Comment: “From an implementation standpoint, if it is possible to convert the excel sheet into an online calculator it would be easier for providers to make more individualized decisions for patients. Ideally, it could be incorporated into the FoxChase (https://cancernomograms.com/nomograms/) nomogram site where the ASSURE RCC calculator also lives (as listed in the introduction). That would could allow for seamless integration of available data from ASSURE RCC into the treatment effect calculator.”
Author response: We very much agree and a key goal of our paper is to exactly motivate such efforts. Following publication of the manuscript, we intent to communicate with various stakeholders in academia, industry, and regulatory agencies to determine ways to develop such web-based tools across different diseases and clinical scenarios.
Reviewer 3 Report
In this article, the authors have described the usage of causal diagrams for making decisions for the treatment of patients. The authors have nicely described how the conventional regression model and historical randomized clinical trial (RCT) can be used to estimate treatment effects. The authors used an adjuvant treatment of renal cell carcinoma as an example to illustrate how to construct causal diagrams and apply them to guide clinical decisions
Author Response
Comment: “In this article, the authors have described the usage of causal diagrams for making decisions for the treatment of patients. The authors have nicely described how the conventional regression model and historical randomized clinical trial (RCT) can be used to estimate treatment effects. The authors used an adjuvant treatment of renal cell carcinoma as an example to illustrate how to construct causal diagrams and apply them to guide clinical decisions.”
Author response: We thank the reviewer for their feedback. We hope that the manuscript will be of help to clinicians and researchers interested in the application of causal diagrams.